# Histone H3.3 sub-variant H3mm7 is required for normal skeletal muscle regeneration

Akihito Harada[1], Kazumitsu Maehara[1], Yusuke Ono[2], Hiroyuki Taguchi[3], Kiyoshi Yoshioka[2], Yasuo Kitajima[2], Yan Xie[3], Yuko Sato[4], Takeshi Iwasaki[1], Jumpei Nogami[1], Seiji Okada[5], Tetsuro Komatsu[1], Yuichiro Semba[1], Tatsuya Takemoto[6], Hiroshi Kimura[4], Hitoshi Kurumizaka [3] & Yasuyuki Ohkawa [1]

Regulation of gene expression requires selective incorporation of histone H3 variant H3.3 into chromatin. Histone H3.3 has several subsidiary variants but their functions are unclear. Here we characterize the function of histone H3.3 sub-variant, H3mm7, which is expressed in skeletal muscle satellite cells. H3mm7 knockout mice demonstrate an essential role of H3mm7 in skeletal muscle regeneration. Chromatin analysis reveals that H3mm7 facilitates transcription by forming an open chromatin structure around promoter regions including those of myogenic genes. The crystal structure of the nucleosome containing H3mm7 reveals that, unlike the S57 residue of other H3 proteins, the H3mm7-specific A57 residue cannot form a hydrogen bond with the R40 residue of the cognate H4 molecule. Consequently, the H3mm7 nucleosome is unstable in vitro and exhibited higher mobility in vivo compared with the H3.3 nucleosome. We conclude that the unstable H3mm7 nucleosome may be required for proper skeletal muscle differentiation.

[1] Division of Transcriptomics, Medical Institute of Bioregulation, Kyushu University, 3-1-1 Maidashi, Higashi, Fukuoka 812-0054, Japan. [2] Graduate School of Biomedical Science, Nagasaki University, 1-12-4 Sakamoto, Nagasaki 852-8523, Japan. [3] Laboratory of Structural Biology, Graduate School of Advanced Science and Engineering, Research Institute for Science and Engineering, and Institute for Medical-oriented Structural Biology, Waseda University, 2-2 Wakamatsu-cho, Shinjuku-ku, Tokyo 162-8480, Japan. [4] Cell Biology Center, Institute of Innovative Research, Tokyo Institute of Technology, 4259 Nagatsuta, Midori-ku, Yokohama, Yokohama 226-8503, Japan. [5] Department of Advanced Medical Initiatives, Graduate School of Medical Sciences, Kyushu University, Fukuoka 812-8582, Japan. [6] Institute of Advanced Medical Sciences, Tokushima University, 3-18-15 Kuramoto-cho, Tokushima 770-8503, Japan. These authors contributed equally: Akihito Harada, Kazumitsu Maehara. Correspondence and requests for materials should be addressed to Y.O. (email: yohkawa@bioreg.kyushu-u.ac.jp)

Chromatin, composed of genomic DNA and histone octamers, dynamically changes its structure to achieve correct gene regulation, replication, and repair of genomic DNA. Thus far, chromatin structures have been characterized by analysis of nucleosomes. Change in nucleosome positioning is required for RNA polymerase II recruitment onto transcription start sites (TSSs). The recent advances in understanding chromatin structure have unveiled the diversity of histone proteins, the so-called histone variants, which play specific roles in specialized local chromatin structures[1]. However, the functions of each histone variant are still unclear because the histone variants are diverse.

Many histone H3 variants have been studied in the past decade. CenH3/CENP-A has been characterized by its function in the centromere to promote kinetochore functions[2,3]. It is overexpressed in cancer cells and mislocalized to chromosome arms forming heterotypic nucleosomes with the histone variant H3.3[4]. The diversified function of H3.3 has also been studied in active transcribed gene loci during cell differentiation or development, as well as in the centromere[5]. For example, H3.3 knockout (KO) results in embryonic lethality in mice[6], and maternal H3.3 is required for the expression of pluripotency genes in oocyte reprogramming[7]. Myogenesis is a well-characterized model to understand H3.3 function in gene regulation upon skeletal muscle differentiation. H3.3 is incorporated into skeletal muscle gene promoters prior to differentiation[8,9]. The selective incorporation of H3.3 is antagonized by the ectopic expression of canonical histone H3.1, which results in deficient skeletal muscle differentiation[10]. The replacement of canonical histones by variants is considered to have a general and important role in lineage specification[5].

The distinct function of histone H3 variants has also been demonstrated by their involvement in oncogenesis. Mutations in canonical histone H3.1 are involved in diffuse intrinsic pontine gliomas, while mutation of H3.3 promotes glioblastoma[11] and is reported in chondroblastoma and giant cell tumors[12,13]. These reports indicate that the selective function of histone H3 variants is required for gene expression as well as development.

We recently identified 14 novel mouse histone H3 variants termed H3mm6-18 and H3t, some of which showed tissue-specific expression[14]. Thirteen of these variants share functional amino acid motifs with H3.3 and are suggested to be subsidiary variants (sub-variants) of H3.3. We also identified an H3.1 sub-variant, H3t; the H3t mRNA contains a stem-loop RNA structure after the coding region[15] indicating that it is a replication-coupled histone. Deletion of the testis-specific H3t gene causes male sterility[16]. The H3mm6–18 genes are predicted to be replication-independent histone variants, because they have polyadenylation signal sequences and are located outside the histone clusters. However, the unique function of each H3.3-type variant remains unclear. The ectopic expression of H3mm7 and H3mm11 resulted in specific expression profiles in skeletal muscle tissues that were regulated during the differentiation of C2C12 mouse myoblast cells. We therefore hypothesized that these histone variants regulate gene expression during lineage commitment by forming a special chromatin structure.

Here we demonstrate that the histone variant H3mm7 is required for skeletal muscle differentiation. H3mm7 is expressed in early stage muscle stem cells, and deletion of the H3mm7 gene leads to deficient muscle cell regeneration and differentiation. Transcriptome analysis of H3mm7$^{-/-}$ cells reveals rate changes in the expression levels of skeletal-muscle-related genes. Structural and biochemical analyses indicates that H3mm7 incorporation renders the nucleosome unstable; the H3mm7-specific A57 residue is unable to form a hydrogen bond with the cognate H4 molecule. H3mm7 may enhance the levels of activated gene expression during differentiation by forming a relaxed chromatin structure.

## Results

**H3mm7 is expressed in skeletal muscle tissues.** We previously reported that the histone variant H3mm7 is expressed in mouse skeletal muscle tissues and its forced expression positively enhances skeletal muscle differentiation in C2C12 cells[14]. We first attempted to identify which cells in skeletal muscle tissues express histone H3 variants. Mature skeletal muscle tissue is composed of myofibers (MFs) and a small number of satellite cells (SCs), blood vessels, and nerve tissues. Hence, global gene expression analysis using CEL-Seq2[17] was performed at different stages of myogenesis. We prepared SCs by muscle injury induced by cardiotoxin (CTX) injection into the gastrocnemius muscle in Pax7-CreERT2; Rosa26-YFP mice, and then Pax7 (YFP)-expressing cells were isolated as SCs (quiescent SCs from uninjured muscle and activated SCs from injured muscle). MFs were isolated from the extensor digitorum longus (EDL) muscle (see Methods for details). We found that H3mm7 was preferentially expressed in SCs (activated: Pax7+, Myod1+ cells and quiescent: Pax7+, Calcr+ cells)[18] compared with MFs (Fig. 1a and Supplementary Fig. 1 for the correlation coefficient matrix of each sample). We additionally confirmed the SC-specific expression of H3mm7 using variant-specific reverse transcriptase quantitative polymerase chain reaction (RT-qPCR) Supplementary Fig. 2a), as used for H3.3-coding genes (H3f3a, H3f3b) (Supplementary Fig. 2b). H3mm7 showed decreased expression in MFs ($p = 0.044$); therefore, we considered H3mm7 to function in earlier stages of skeletal muscle regeneration.

To evaluate the physiological function of H3mm7 in SCs during muscle regeneration, we generated a H3mm7 KO mouse line by introducing Cas9 protein and clustered regularly interspaced short palindromic repeat (CRISPR)-derived RNA (crRNA)/trans-activating crRNA (tracrRNA) into in vitro fertilized zygotes by electroporation[19] (Supplementary Fig. 3). As expected, because both male and female H3mm7$^{-/-}$ mice were viable and indistinguishable from control mice, H3mm7 was not necessary during development. Immunostaining for developmental myosin heavy chain (dMHC) in the tibialis anterior (TA) muscle from H3mm7$^{-/-}$ mice was performed at 14 days after CTX injection (Fig. 1b). dMHC is transiently expressed in regenerating fibers in the early phase but is downregulated in the late phase of muscle regeneration[20,21]. Some dMHC-positive immature regenerating fibers and a substantial number of thin fibers were observed in H3mm7$^{-/-}$ TA muscle (Fig. 1b, c), indicating that muscle regeneration was delayed in H3mm7$^{-/-}$ mice.

We assessed whether the delay in muscle regeneration in H3mm7 KO mice was caused by a decrease in the number of SCs. Immunohistochemistry on day 5 after the induction of muscle injury by CTX showed no notable change in the proportions of Pax7(+)MyoD(−), Pax7(+)MyoD(+), and Pax7(−)MyoD(+) cells around the regenerated muscle (Supplementary Fig. 4). We next assessed changes in the gene expression profile upon KO by RNA-seq (CEL-Seq2) using tissue sections. We fitted the gene expression profile data under injured and uninjured conditions at days 5 and 14 into a statistical model using the DESeq2 software (Supplementary Table 1). We found that the injury-induced changes to expression levels of SC-related genes were analogous between H3mm7$^{+/+}$ and H3mm7$^{-/-}$ mice. Of these SC-related genes with high fold changes in Injured/Uninjured, Pax7 demonstrated a $\log_2$ fold change = 3.19 and its expression level was 9-fold higher in the Injured condition on days 5 and 14. Ncam1 similarly showed an 18-fold higher expression level in the

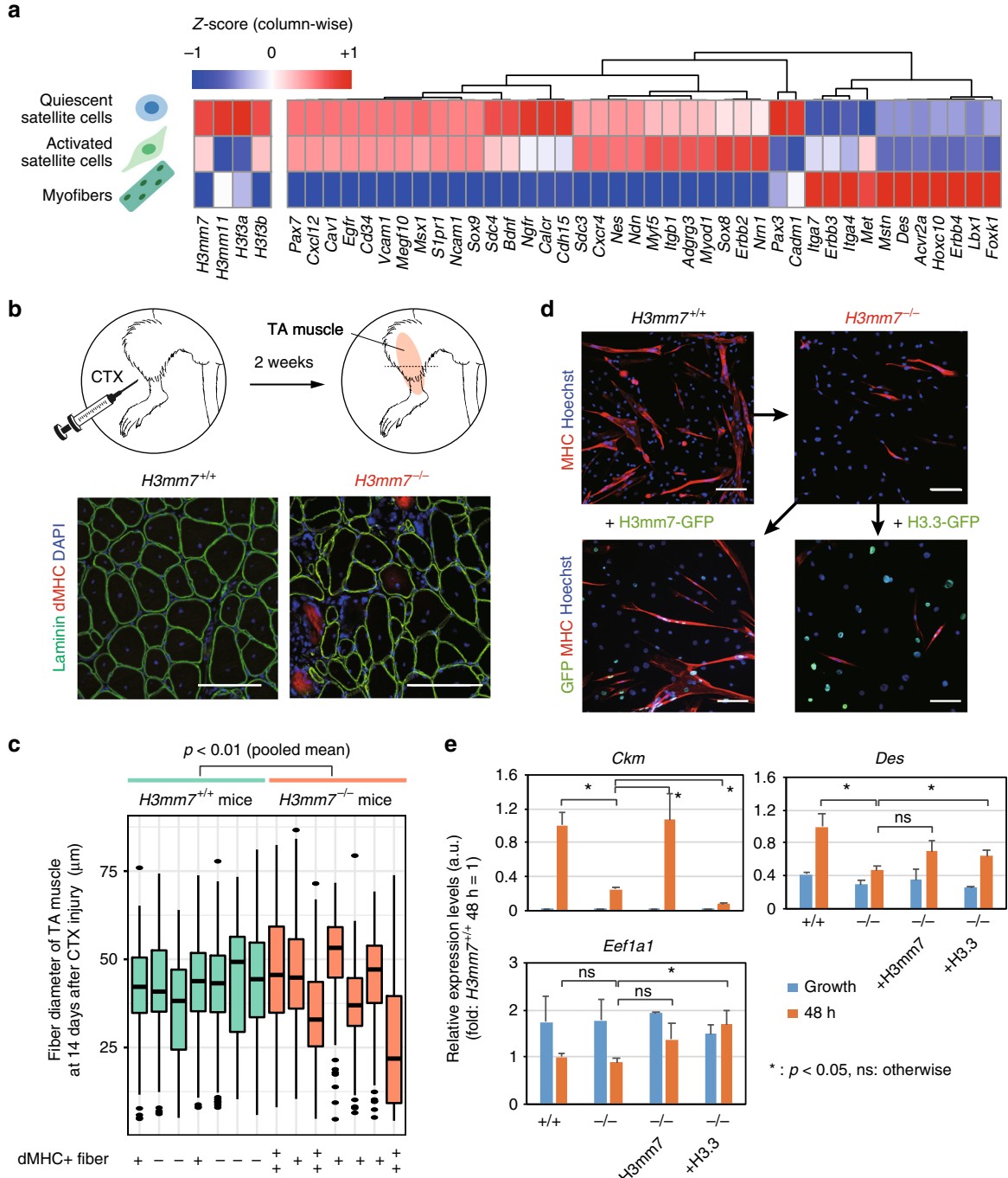

**Fig. 1** H3mm7 is required for normal skeletal muscle regeneration and differentiation. **a** The *H3mm7* gene is expressed in satellite cells. The colors of the tiles indicate the average expression level (*n* = 3) of H3 variants, satellite cell marker genes, and differentiation markers[45] as the column-wise relative expression levels (*Z*-score). **b** Some of the fibers showed incomplete muscle regeneration in *H3mm7⁻/⁻* mice. Immunostaining of a regeneration marker (dMHC; red) and laminin (green) in TA muscle at 14 days after CTX injection. **c** CSA measurements of *H3mm7⁻/⁻* TA muscle fibers showed thinner fiber formation during muscle regeneration. The approximated fiber diameters of *H3mm7⁺/⁺* and *H3mm7⁻/⁻* mice are shown as box plots (*n* = 177, 175, 231, 161, 179, 154, 156, 132, 153, 251, 138, 192, 141, 339 cells for individuals, respectively). The signs at the bottom indicate the observed number of dMHC-positive fibers (−: none, +: at least one, ++: >4). The medians, 25/75th percentiles, and 1.5 IQR (inter-quartile range) are employed to draw the box plots. **d** *H3mm7⁻/⁻* cells showed SK muscle differentiation deficiency in C2C12 cells and an H3mm7-rescued clone recovered SK muscle differentiation. Immunostaining of markers of fiber formation (MHC; red), rescued histone variants (GFP; green), and the nucleus (Hoechst; blue) in *H3mm7⁻/⁻* and *H3mm7⁺/⁺* C2C12 cells are shown. **e** The loss of skeletal muscle marker gene expression levels was partially rescued by the H3mm7 expression. The expression levels relative to *H3mm7⁺/⁺* at 48 h after differentiation stimuli were measured by RT-qPCR (*n* = 3). The error bars indicate ±1 SD. Two-sided Welch's t-test was performed and the *p*-values are *Ckm*: 0.001, 0.042, 0.003, *Des*: 0.002, 0.064, 0.032 and *Eef1a1*: 0.159, 0.126, 0.031 (*H3mm7⁺/⁺* vs. *H3mm7⁻/⁻*, *H3mm7⁻/⁻* vs. +*H3mm7*, *H3mm7⁻/⁻* vs. +H3.3 respectively). The scale bars in the immunostaining images are 100 µm

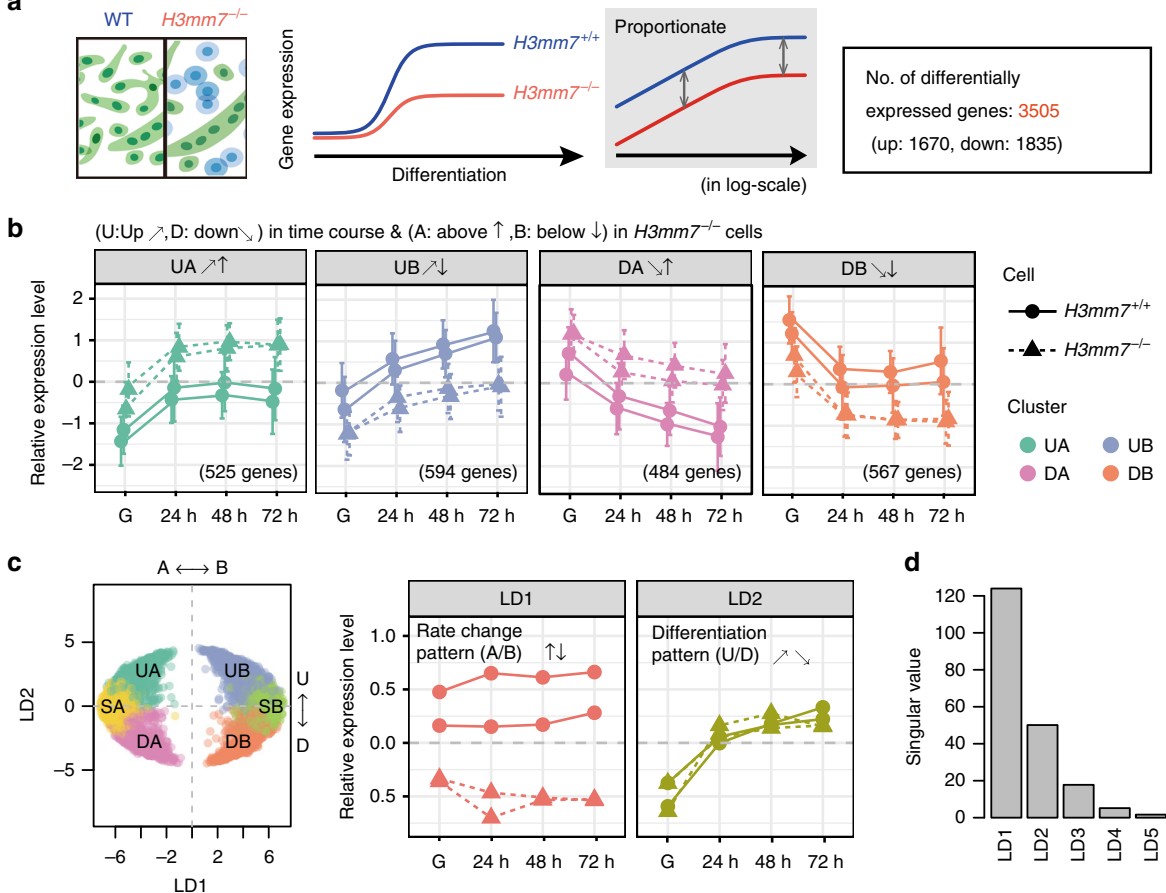

**Fig. 2** $H3mm7^{-/-}$ causes the rate change of gene expression during differentiation. **a** For all DEGs, the rate change of gene expression was observed throughout the differentiation time course. The illustrations explain the result of the statistical test against RNA-seq data. **b** The four types of rate changing patterns during differentiation are explained by the combination of the patterns: Up/Down in time and Above/Below in $H3mm7^{-/-}$ cells. The points indicate the cluster average of the relative (gene-wise $Z$-normalized) expression levels of two cell types ($H3mm7^{+/+}$ and $H3mm7^{-/-}$) with two replicates. The error bars indicate ± 1 SD. The numbers of the genes in the clusters are indicated in parentheses. **c** LDA extracted two major gene expression patterns during differentiation that correspond to A/B and U/D. The points indicate the genes of the clusters in LD space spanned by LD1 (middle) and LD2 (right) components. **d** The singular values of the LDA (the ratio of between- and within-cluster standard deviations) indicate that the two most relevant components to discriminate the clusters were LD1 (A/B) and LD2 (U/D)

*Injured* condition (log$_2$ fold change = 4.19) and the interaction term for *Injured* and *Day 14* was −4.16 (one eighteenth), i.e., the expression level returned to the level in uninjured muscle on day 14. We further observed that downregulated genes in $H3mm7^{-/-}$ mice included mature muscle fiber-related genes, such as *Tnnc2*[22,23]. These results suggest that the loss of *H3mm7* might affect the differentiation process, while leaving the population of SCs unchanged. We further evaluated the function of H3mm7 in isogenic C2C12 cells, which express H3mm7 prior to differentiation stimuli and are used as a differentiation model of myogenesis[24]. Two independent clones of *H3mm7* homozygous KO ($H3mm7^{-/-}$) C2C12 cells were constructed using the CRISPR/Cas9-system (see Supplementary Fig. 5a, b for the deletion strategy of *H3mm7* in tetraploid C2C12 cells). Immunostaining of the skeletal muscle marker MHC showed impaired differentiation in $H3mm7^{-/-}$ cells (Fig. 1d). Furthermore, the ectopic expression of H3mm7-GFP, but not H3.3-GFP, rescued the expression level of MHC in $H3mm7^{-/-}$ cells (Fig. 1d, lower panel). The fusion index for the time course of differentiation showed that the formation of multinucleated MFs in H3mm7-rescued cells was comparable to that in wild-type (WT) ($H3mm7^{+/+}$) cells, while H3.3-GFP decreased the fusion index in $H3mm7^{-/-}$ cells (Supplementary Fig. 6). We also

evaluated the total amount of histone H3 in C2C12 cells using an antibody that recognizes the C-terminus of histone H3 (anti-H3-C; see Methods for details) and found no significant difference in the amounts present (Supplementary Fig. 7a), while the soluble H3-C level was decreased only in H3mm7$^{-/-}$ rescued by H3mm7-GFP cells (Supplementary Fig. 7b). The expression of exogenous green fluorescent protein (GFP)-fused histones did not affect the endogenous H3 histones. Therefore, the impaired differentiation in $H3mm7^{-/-}$ cells was not caused by insufficient amounts of histone H3.

We also confirmed the result in terms of mRNA levels using the representative skeletal muscle markers, *Ckm* and *Des*, and the housekeeping gene, *Eef1a1*, as a control (Fig. 1e). The mRNA level of *Ckm*, which is highly expressed in differentiation, was lower in $H3mm7^{-/-}$ cells than in $H3mm7^{+/+}$ cells at 48 h after the differentiation stimulus ($p = 0.001$; lower than $H3mm7^{+/+}$) and was rescued by H3mm7 expression ($p = 0.042$; higher than $H3mm7^{-/-}$). H3.3 did not rescue the expression levels of *Ckm* at 48 h but rather caused a decrease in expression levels of some myogenic genes ($p = 0.003$; lower than $H3mm7^{-/-}$). Other skeletal muscle-related genes highly expressed after differentiation were also rescued by H3mm7 (Supplementary Fig. 8). However, the level of *Des*, expressed prior to differentiation, was

**Table 1 Top-enriched Gene ontology terms using GSEA for 3505 DEGs for *H3mm7*$^{-/-}$ vs. *H3mm7*$^{+/+}$ C2C12 cells**

| Downregulated GO terms (top-15) | NES[a] | Upregulated GO terms (top-15) | NES[a] |
|---|---|---|---|
| Ribosome biogenesis (GO:0042254) | −2.666 | Spleen development (GO:0048536) | 2.562 |
| Ribosome assembly (GO:0042255) | −2.657 | Cilium assembly (GO:0042384) | 2.446 |
| rRNA processing (GO:0006364) | −2.645 | Cilium morphogenesis (GO:0060271) | 2.333 |
| Translation (GO:0006412) | −2.631 | Cilium organization (GO:0044782) | 2.304 |
| Peptide biosynthetic process (GO:0043043) | −2.59 | Positive regulation of defense response to virus by host (GO:0002230) | 2.289 |
| rRNA metabolic process (GO:0016072) | −2.581 | Regulation of defense response to virus (GO:0050688) | 2.259 |
| Amide biosynthetic process (GO:0043604) | −2.566 | Cellular component assembly involved in morphogenesis (GO:0010927) | 2.238 |
| Ribosomal small subunit assembly (GO:0000028) | −2.555 | Negative regulation of transforming growth factor beta receptor signaling pathway (GO:0030512) | 2.233 |
| Ribosomal small subunit biogenesis (GO:0042274) | −2.553 | Negative regulation of cellular response to transforming growth factor beta stimulus (GO:1903845) | 2.233 |
| Ribosomal large subunit biogenesis (GO:0042273) | −2.5 | Positive regulation of calcium ion transport (GO:0051928) | 2.206 |
| ncRNA processing (GO:0034470) | −2.486 | Negative regulation of vasculature development (GO:1901343) | 2.201 |
| Organonitrogen compound biosynthetic process (GO:1901566) | −2.441 | Negative regulation of cellular response to growth factor stimulus (GO:0090288) | 2.201 |
| Peptide metabolic process (GO:0006518) | −2.412 | Regulation of defense response to virus by host (GO:0050691) | 2.194 |
| Skeletal muscle cell differentiation (GO:0035914) | −2.4 | Cell projection morphogenesis (GO:0048858) | 2.163 |
| Regulation of myotube differentiation (GO:0010830) | −2.372 | Xenophagy (GO:0098792) | 2.153 |

[a] NES normalized enrichment score

rescued by H3.3 ($p = 0.032$), which was not the case for H3mm7 ($p = 0.064$). The levels of the control (housekeeping) gene *Eef1a1* were not significantly changed in *H3mm7*$^{+/+}$, *H3mm7*$^{-/-}$, or the rescued clones of H3mm7, although an elevated level of *Eef1a1* was observed upon H3.3-GFP expression. Taken together, while not all genes that were downregulated by the loss of the H3mm7 gene were rescued by the ectopic expression of H3mm7-GFP, a group of myogenesis-related genes were observed to be significantly rescued (Fig. 1e and Supplementary Fig. 8), suggesting that H3mm7 might have a function to regulate these genes. These results suggest that the expression of H3mm7 might affect upregulated genes in differentiation.

***H3mm7*$^{-/-}$ causes rate changes in gene expression levels**. Next, to identify genes where expression levels were altered by H3mm7 during differentiation, we performed time-course RNA-seq analysis with C2C12 cells. We collected C2C12 cells at different growth states 24, 48, and 72 h after initiation of the differentiation stimulus.

We attempted to detect differentially expressed genes (DEGs) that had lost the kinetics of normal myogenesis during differentiation. Loss of gene expression kinetics is commonly detected in the gene KO or knockdown models as representatively shown in Supplementary Figs. 9 and 10 for si-mH2A1.2 in C2C12 cells[25] and for BMAL KO in hip cartilage samples[26], respectively. In the latter study, the authors utilize DESeq2's likelihood ratio test in a set-up analogous to ours (Time and Phenotype factors) to demonstrate that a group of genes lose the periodical pattern of expression. Following their method, we employed a statistical model using DESeq2 (a graphical explanation is provided in Supplementary Fig. 11): Gene expression levels at time $t$ in KO cells = (WT expression level at time $t$) + (KO effect through time) + (extra effect at time $t$ in KO cells). However, the likelihood ratio test of the model revealed that no genes demonstrated a loss at any time point. Therefore, the results indicated that no genes showed a complete loss or abnormal activation of expression induced by loss of H3mm7 during differentiation. The secondary finding from the statistical test was that, for all DEGs, the change caused by *H3mm7* KO could be sufficiently explained by the "KO effect through time" term above; i.e., the effect of *H3mm7* KO was viewed as a constant rate change (offset in log-scale) through time (Fig. 2a, Supplementary

Fig. 11). Overall, 3505 genes were detected as DEGs in *H3mm7*$^{-/-}$ cells. We therefore hypothesized that the loss of H3mm7 caused the expression rate change of differentiation-related genes and thus the proportion of successfully differentiated cells decreased despite the isogenicity of the *H3mm7*$^{-/-}$ C2C12 cells.

To characterize these 3505 genes, we performed Gene Set Enrichment Analysis (GSEA) (see Table 1 and Supplementary Table 2 for detailed lists of the GSEA analysis results) to extract H3mm7-dependent gene sets. As expected, negative enrichment was observed in skeletal muscle cell differentiation (GO:0035914; normalized enrichment score (NES) = −2.40). In addition, increased inhibition of the response to the growth factor, which is considered a reaction during skeletal muscle differentiation, was also observed (GO:0090288; NES = 2.20). However, myogenesis-independent gene sets were also enriched both in the downregulated genes, for example, ribosome biogenesis (GO:0042255, NES = −2.67), and the upregulated genes, such as cilium assembly (GO:0042384, NES = 2.45). These results suggest that, while H3mm7 is not involved specifically in the expression of genes related to myogenic differentiation, it may be required to upregulate gene expression upon differentiation.

To determine the temporal expression patterns of the genes responsible for the rate change phenomenon during differentiation, we classified the 3505 genes into categories according to gene expression patterns in time. The $k$-means clustering of the time-course RNA-seq data resulted in four types of gene expression patterns that could be explained by the combination of the upregulation/downregulation (abbreviations: U, D) in the course of differentiation and the above/below (A, B) levels of gene expression in *H3mm7*$^{-/-}$ cells compared with *H3mm7*$^{+/+}$ cells (Fig. 2b). Flat patterns (no significant change in time; cluster SA and SB) were also observed but were not considered the result of *H3mm7*$^{-/-}$, because DESeq2 only captured the genes that varied between the replicates of *H3mm7*$^{+/+}$ cells (Supplementary Fig. 12). These data indicated that loss of the *H3mm7* gene caused the rate change in expression of upregulated and downregulated genes upon differentiation.

We further confirmed that the rate change had occurred with upregulation/downregulation of gene expression after the loss of *H3mm7* by Fisher's linear discriminant analysis (LDA). LDA was used to estimate the superpositions of the expression patterns in the clusters of Fig. 2b (Fig. 2c). The singular values (Fig. 2d) representing the performance of the class separation showed the

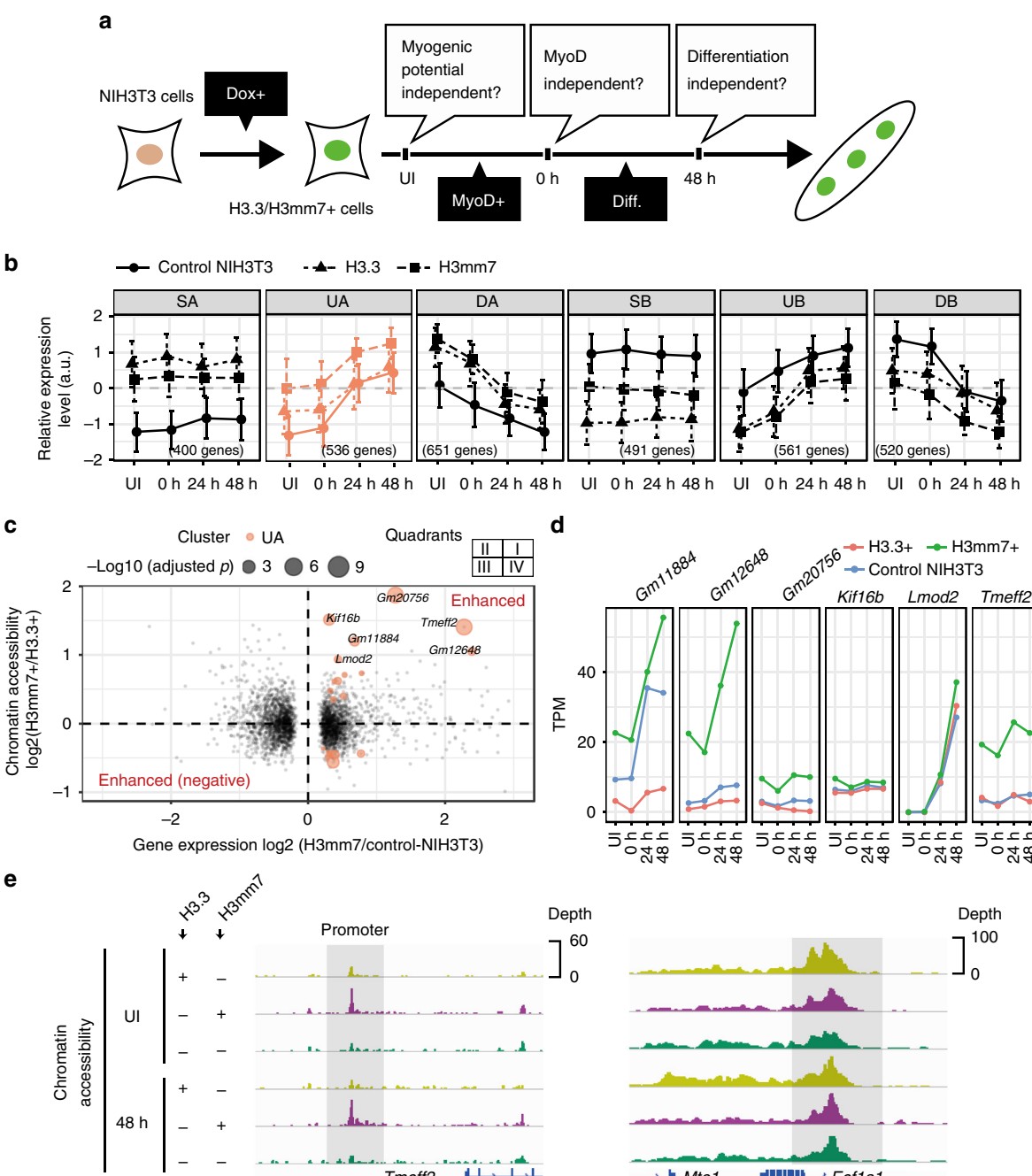

**Fig. 3** The incorporation of H3mm7 enhances activated gene expression by increasing chromatin accessibility. **a** Trans-differentiation model by MyoD infection using H3mm7/H3.3-GFP-expressing NIH3T3 cells. This model was utilized to confirm each dependency for exerting the function of H3mm7 on the inherent cell lineage (skeletal muscle potential), transcription factor (MyoD), and differentiation stimuli (serum starvation). **b** The six types of gene expression patterns in time-series RNA-seq data of NIH3T3 cells. The plot was created as in Fig. 2b. The error bars indicate ±1 SD of the genes (*n*: the number of genes in the clusters) **c** Marked enhancement of gene expression occurred in activated genes during differentiation (cluster UA). The scatter plot of log2-transformed fold change of gene expression (H3mm7+ vs. WT cells, through time) and chromatin accessibility changes (H3mm7+ vs. H3.3+ cells, through time) are shown. The colors correspond to the clusters in **b**. The sizes of the points are proportional to −log10 (adjusted *p*-values of chromatin accessibility change). Enhanced: upregulated genes in control NIH3T3 cells over time with higher expression levels and higher accessibility in H3mm7+ cells compared with H3.3+ cells. Enhanced (negative): downregulated genes in control NIH3T3 cells over time with lower expression levels and lower accessibility in H3mm7+ cells compared with H3.3+ cells. **d** The expression levels of genes that showed marked enhancement. The expression levels are calculated as transcripts per million (TPM). **e** IGV snapshot of ATAC-seq shows increased chromatin accessibility at an H3mm7 incorporated promoter

two major components LD1 (first component of LDA; corresponding to A and B) and LD2 (U, D) in contrast to the other minor ones (LD3, 4, and 5). The singular values of LD1 and LD2 were 123.98 and 50.07 (84.3% and 13.8% of total between-class variance), respectively. This suggested that the rate change (bias) between *H3mm7*$^{-/-}$ and *H3mm7*$^{+/+}$, and subsequently the upregulation/downregulation in differentiation, were indeed the two major components that explain the result of *k*-means clustering.

**H3mm7 functions independently of muscle differentiation.** We found that knocking out the *H3mm7* gene led to a rate change in gene expression levels. However, it remains to be determined whether H3mm7 functions independently upon incorporation or only in concert with other factors. We confirmed that NIH3T3 fibroblast cells naturally lack the *H3mm7* gene locus by analysis of amplicon sequences (Supplementary Fig. 13). Hence, we performed time-course RNA-seq in NIH3T3 cells with GFP-fused H3mm7 and with H3.3 as control. In this experiment, if the rate change takes place after differentiation induction by MyoD infection, it is transcription-factor-dependent. However, if it occurs only upon over-expression, it is dependent on histone variant incorporation (Fig. 3a).

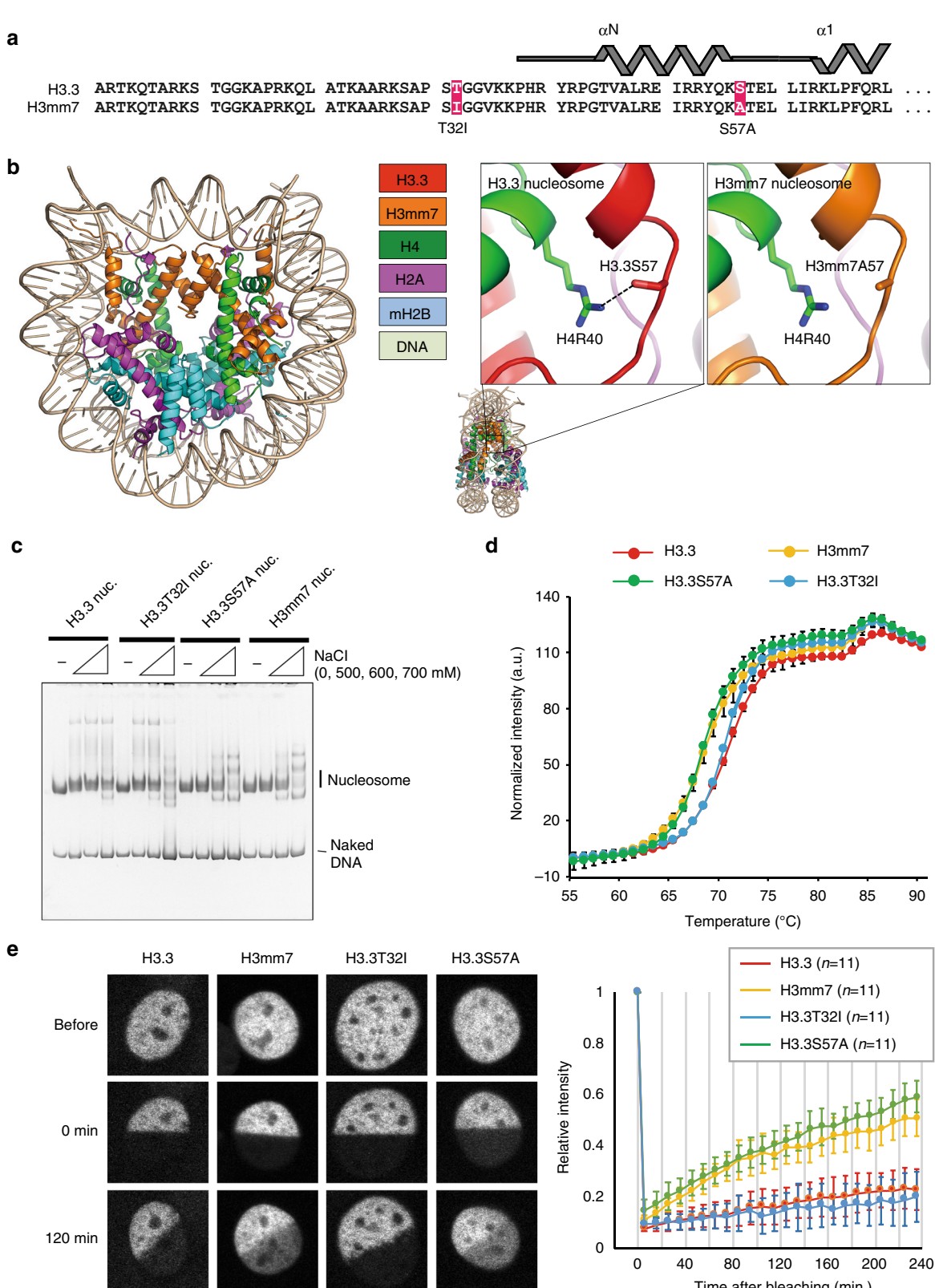

As a result of $k$-means clustering of time-series RNA-seq data, we obtained six clusters (U, D, S) × (A, B) for NIH3T3 cells (Fig. 3b). The rate change in gene expression levels was observed at all time points before infection (UI; uninfected) to 48 h after differentiation induction. This indicates that the function of H3mm7 is independent of MyoD infection and of differentiation stimuli. Moreover, we determined that the expression of H3mm7 led to a greater rate change in expression levels compared with that caused by H3.3 (Supplementary Fig. 14a). We also performed an analysis analogous to that in Supplementary Fig. 14a against control (Supplementary Fig. 14b). The results showed that both H3mm7 and H3.3 cases demonstrated significant mean differences against WT in all the clusters, which suggests that the ectopic expression of H3.3 and H3mm7 have analogous effects on the gene expression profile. However, the difference was not significant at 24 h in the clusters UA and DA in the case of H3.3, and H3mm7 induced greater rate changes in comparison with H3.3; these facts might lead to the H3mm7-specific gene expression profile.

GSEA analysis also showed a consistency of the C2C12 data (Supplementary Table 3): that a rate change phenomenon was observed in genes other than those related to skeletal muscle differentiation (GO:1901861, Regulation of muscle tissue development, NES = 2.07). We confirmed RNA-seq data by qPCR for the representative differentiation marker genes belonging to cluster UA (Supplementary Fig. 15). While the plot in Fig. 3b showed notable rate changes in differentiation induced by H3mm7, the genes in Supplementary Fig. 15 included those for which the enhancement effect was greater in H3mm7 (*Des* and *Tnnt2*), those in which H3.3 and H3mm7 have comparable enhancement effects (*Acta1*, *Ckm* and *Tnni2*), and a group of genes including *Myh3* for which the ectopic expression of H3.3 resulted in lower expression levels compared with WT. The results in Fig. 1d and Supplementary Fig. 6 show that H3.3 only partially compensated the function of H3mm7; however, the results of RNA-seq analysis in NIH3T3 cells also indicate that H3mm7 has a greater impact on gene expression than H3.3.

**H3mm7 enhances gene expression via accessible chromatin**. We investigated whether the H3mm7-specific function occurs only after selective recruitment to specific regions induced by recruitment factors, such as chaperones specific to H3mm7. First, we evaluated whether there are unique H3mm7 incorporation sites in NIH3T3 cells ectopically expressing GFP-tagged histone H3 variants. Then we evaluated histone modifications (H3K4me3 and H3K27me3) at the sites of GFP-tagged histone incorporation as well as chromatin accessibility by assay for transposase-accessible chromatin using sequencing (ATAC-seq). Both GFP-histones H3.3 and H3mm7 were preferentially incorporated in promoter regions (Supplementary Fig. 16 for GFP peak annotations) and did not show a substantial difference in the incorporated sites on the genome. H3K4me3 and H3K27me3

were not affected either in signal levels or their distribution pattern on the genome by the ectopic expression of GFP-histones H3.3 and H3mm7 compared with control NIH3T3 cells (Supplementary Figs. 17 and 18). Therefore, this indicates that the regulation of expression rate by H3mm7 incorporation did not originate from the specific change of histone modification or incorporation sites on the genome. Despite the fact that H3.3 and H3mm7 are nearly identical in their sequences, localization, and histone modification preferences, the rate change effect of H3mm7 made a notable difference in the gene expression profile after differentiation.

We, therefore, focused on genes whose chromatin structures were altered as a result of the forced expression of H3mm7. First, we identified the group of genes with increased accessibility to H3mm7 vs. control NIH3T3 cells by evaluating the ATAC-seq signal levels in the promoter regions (Fig. 3c). H3mm7-induced DEG groups, classified in Fig. 3b, were used to determine groups that caused a change in chromatin accessibility. As in the case of the gene expression profiles considered above, the mean accessibility levels were already high at the UI stage when the variant is incorporated, and the high accessibility levels were maintained after differentiation (Supplementary Fig. 19). Then we evaluated the relationship between the fold change in accessibility (H3mm7+ cells/H3.3+ cells) and the fold change in gene expression (H3mm7+ cells/WT) by scatter plot (Fig. 3c). The enhancement effect of H3mm7 on gene expression was greater than that of H3.3 (Fig. 3b and Supplementary Fig. 14); therefore, we compared the numbers of genes belonging to cluster UA in quadrants I and IV (i.e., greater enhancement effect both in chromatin accessibility and gene expression in H3mm7+ cells vs. greater enhancement effect in H3.3+ cells). We found the result to be I: 39 vs. IV: 9, and hence we confirmed that there were more genes with enhancement levels greater by >2 fold changes in chromatin accessibility and gene expression induced by the ectopic expression of H3mm7 (Supplementary Fig. 20). Therefore, we concluded that genes belonging to cluster UA were enhanced more in both accessibility and expression by H3mm7 than by H3.3. We observed increased expression levels for individual genes after induction of H3mm7 (Fig. 3d). Figure 3e shows increased accessibility of the representative *Tmeff2* gene promoter upon H3mm7 induction (2.66 fold change compared with H3.3+ cells; adjusted $p = 4.07 \times 10^{-12}$) while the *Eef1a1* promoter showed no difference between H3.3- and H3mm7-incorporated promoters (adjusted $p = 0.36$). We additionally performed ATAC-seq analysis using WT and H3mm7-KO C2C12 cells (Supplementary Fig. 21). The data demonstrated that loss of H3mm7 caused a decrease in chromatin accessibility (chromatin accessibility up: 25 genes, down: 137 genes), while the expression levels of the affected genes were decreased only after differentiation stimuli (Supplementary Fig. 22). These results suggest that H3mm7 is expressed prior to differentiation and supports the differentiation process. Together, these results suggest that the change of chromatin accessibility at

**Fig. 4** Ala57 allows the mobility of the H3mm7 nucleosome. **a** T32I and S57A are only the differences between H3.3 and H3mm7. N-terminals of the amino acid sequence of H3.3 and H3mm7 are shown in the illustration of the secondary structure. **b** High-resolution crystal structure of an H3mm7-containing nucleosome shows the loss of interaction between H3.3S57 and H4R40 in the H3mm7 nucleosome. An overview of the crystal structure of the H3mm7-containing nucleosome (left) and the comparative view around S57 in H3.3 and A57 in H3mm7 (right). **c** H3mm7 nucleosomes were dissociated at a concentration of 600 mM NaCl. The salt-titration assay of H3mm7, H3.3, and the mutant (H3.3S57A and T32I) containing nucleosomes is shown. The positions of the bands that correspond with the intact nucleosome core particle and naked DNA are indicated on the right. **d** H3.3S57A and H3mm7 dissociated from the H2A:H2B dimer faster than the other H3 variant-containing nucleosomes. Thermal stability assay of H3mm7, H3.3, and the point mutants H3.3T32I and H3.3S57A. The lines show the fluorescent intensity that indicates nucleosome dissociation at each temperature. The error bars are ±1 SD ($n = 4$). **e** H3mm7 and H3.3S57A showed a fast histone turnover. FRAP analysis of nucleosomes containing GFP-fused H3mm7, H3.3, and the point mutants. Representative confocal images of FRAP analysis in each H3 variant nucleosome is shown on the left. Plot of average GFP fluorescent recovery rate at each time point after photobleaching is shown on the right. The error bars are ±1 SD. The number of replicates is indicated in parentheses

H3mm7-incorporated genes enhances the upregulation of the expression of myogenic-differentiation-related genes notably in cluster UA to a greater degree compared with H3.3+ cells.

**The S57A substitution causes unstable nucleosomes with H3mm7.** H3.3 and H3mm7 differ in their amino acid sequences only at T32I and S57A (Fig. 4a); however, it is not clear how this small difference in the amino acid composition of H3 results in a marked difference in function of the nucleosome. We therefore analyzed the crystal structure of the nucleosome containing H3mm7 (Fig. 4b). The overall structure of the H3mm7 nucleosome is quite similar to that of the H3.3 nucleosome. However, we found that the interaction between H4R40 and H3.3S57 in the H3.3 nucleosome is absent in the H3mm7 nucleosome, because the S57 residue of H3.3 is replaced by alanine in H3mm7. The dissociation of nucleosomes was observed for both H3mm7 and H3.3S57A under conditions of 600 mM NaCl, but this dissociation was not observed in nucleosomes with H3.3 and H3.3T32I. This suggests that S57A incorporation results in unstable nucleosomes (Fig. 4c). A thermal stability assay was then performed to evaluate the stability of variant nucleosomes (Fig. 4d). The first peak of the fluorescence intensity curve reflects H2A–H2B dissociation, and the second peak reflects the dissociation of H3–H4[27]. We found that the H2A–H2B dimer became dissociated at lower temperatures (60–70 °C) in nucleosomes containing H3mm7 and the H3.3S57A mutant compared with those containing H3.3 and the H3.3T32I mutant. These results indicate that H3mm7 nucleosome stability is less than that of the H3.3 nucleosome and that the H3mm7-specific alanine residue at position 57 is fully responsible for reduced stability of the H3mm7 nucleosome.

To investigate the effect of unstable nucleosomes on the stability of histone variant incorporation in vivo, we performed fluorescence recovery after photobleaching (FRAP) (Fig. 4e). After photobleaching, the fluorescence intensity of GFP-H3.3 and GFP-H3.3T32I dropped to ~10% of the original intensity and recovered to ~20% in 240 min. The intensity just after bleaching was slightly higher for GFP-H3mm7 and GFP-H3.3S57A and recovered up to ~50–60%, indicating that these proteins exchange more rapidly than GFP-H3.3 and GFP-H3.3T32I. Therefore, we concluded that H3mm7 is more mobile than H3.3 in cells, probably because of unstable nucleosome formation due to the A57 residue of H3mm7.

To further assess the enhancement of H3.3S57A on gene expression, we performed RNA-seq for NIH3T3 cells ectopically expressing H3.3S57A/T32I mutants. As expected, the S57A mutant enhanced the expression levels of *Myh3*, *Myog*, and *Tnnt2* genes upon differentiation stimuli (representative marker genes of skeletal muscle differentiation, Supplementary Fig. 23), whereas the T32I mutant did not.

## Discussion
Here we found that H3mm7 is substantially expressed in skeletal muscle SCs. The loss of H3mm7 resulted in delayed skeletal muscle regeneration upon CTX injury. These data suggest that, although H3mm7 is the sub-variant of H3.3, its functions are required for proper muscle regeneration via specific chromatin formation.

The detection of these minor histone variants has been impaired by their highly homologous coding sequences (76%–98.5% compared with H3.3)[14]. Therefore, we previously employed 3′-seq[28] to specifically detect the expression levels of novel histone variants in large quantities of RNAs[14]. In the current study, we used CEL-Seq2 to quantify the expression levels of histone variants and confirmed the variant expression in skeletal muscle stem cells and progenitor cells. Although we have focused on H3mm7 expressed in skeletal muscle, these approaches could be applicable to future studies to detect other histone H3 variants in specific tissues. Several KO studies have produced conflicting results. Jang et al. showed that double-KO mice (*H3f3a* and *H3f3b*) resulted in lethality around embryonic stage 3.5 days post coitum[6]; however, in *H3f3b* KO and *H3f3a*-shRNA-suppressed embryonic stem cells, dysfunction of differential potential was observed[29]. It is possible that the function of H3.3 could be disrupted along with the other H3.3 sub-variants because H3.3 sub-variants (H3mm6-H3mm18) share similar shRNA target sequences[8,9].

The absence of H3mm7 might have disrupted the expression of genes, including those related to skeletal muscle, in the cell population. This non-uniformity in gene expression might manifest as the rate change of gene expression seen in the bulk cell RNA-seq data. Future single-cell analysis might determine whether this rate change phenomenon is mediated by single-cell-level expression variations or from the aggregate of the changes in gene expression levels in each cell. For instance, it has been reported that nucleosome positioning in yeast promoters is involved in the regulation of expression noise[30]. Thus the regulation of nucleosome positioning by H3mm7 might result in the rate change of gene expression. Furthermore, some transcription-repressive chromatin regulatory factors regulate the probability of inactivation of gene expression rather than suppressing gene expression levels in the cell population. This suggests that the presence of a variant in the nucleosome that interacts with a chromatin regulatory factor causes variations in the rate parameters ($k_{on}/k_{off}$) for the activation and inactivation of genes[31] in the local area of H3mm7 nucleosome sites. To elucidate this, it is necessary to perform single-cell synthetic biological analysis on chromatin containing the variant.

Histone variants might be another vital factor along with transcription factors in determining unique expression profiles of tissues. In our analysis using KO mice, H3mm7 deficiency did not have a dramatic effect on animal development, although a phenotype in adult muscle regeneration was observed. H3mm7 is poorly conserved across species, even in mammals; however, *H3mm7* corresponds to *H3F3AP3* in humans, which also has the A57 residue. In the H3.3 nucleosome, the S57 residue forms a hydrogen bond with the R40 residue of the cognate H4 molecule. The crystal structure of the H3mm7 nucleosome revealed that the A57 residue of H3mm7 does not affect the nucleosome structure, but the residue is unable to form a hydrogen bond with H4 because of the absence of a hydrogen donor (Fig. 4b). Amino acid conservation of such a key residue may be an important hallmark for identification of functional histone variant homologs across species. In-depth molecular evolutionary analysis of these less-conserved histone variants might further our understanding of histone variants as a driving force in evolution. Another histone variant, *H3mm11*, showed an expression pattern closer to that of *H3f3a* than to that of *H3mm7* (Fig. 1a). GFP-H3mm11 was detected in abundance in TSS and its gene expression profile was observed to change after differentiation[14]. Hence, the structural changes in chromatin manifested through various expression patterns of histone variants might contribute to the transition of chromatin status at cell differentiation.

Because H3mm7 and H3.3 had similar patterns of incorporation and histone modification, the observed difference in their functions might be sought in the difference in their amino-acid residues. For instance, the phosphorylation of H3 T32 has been reported for plant and animal cells[32,33]. Since peptides with H3 T32ph have a slightly increased number of interactions with histone chaperones[34], H3T32I might induce a change in the status of such interactions. H3K56ac is a critical modification for

nucleosome assembly in yeast[35,36] and the S57 residue in H3.3 is replaced by alanine in H3mm7. Phosphorylation of serine 57 promotes a nucleosomal transaction when lysine 56 is acetylated[37]. However, the binary switch model indicates a 'methyl/phos switch' between Lys56 and Ser57[38]. This supports the idea that H3mm7 promotes the function of K56ac. Structural and biochemical analyses demonstrated that H3mm7 nucleosomes are unstable without posttranscriptional modification. The modification of S57A-substituted H3.3 will be studied in future.

## Methods

**Ethical statement**. All animal procedures were conducted in accordance with the Guidelines for the Care and Use of Laboratory Animals and were approved by the Institutional Animal Care and Use Committee (IACUC) at Kyushu University, Nagasaki University and Tokushima University.

**Generation of *H3mm7* KO mice**. H3mm7 KO mice were generated with a genome editing technique as described previously with several modifications[19]. Briefly, in vitro fertilized zygotes (C57BL/6 × C57BL/6) were electroporated with 100 ng/µl Cas9 protein, 100 ng/µl crRNAs, and 100 ng/µl tracrRNA. Two crRNAs targeting upstream and downstream of H3mm7 gene were used. We designed the target sequences of two crRNAs as follows: H3mm7 crRNA1 (5′-TATAGA-TAAAGGAATAACTA-3′), H3mm7 crRNA2 (5′-ACATCACTTATC-TATTGCTT-3′). Electroporated zygotes were transferred into the oviduct of pseudopregnant female mice, and the mutant mice were born on E19. Cas9 protein, crRNAs, and tracrRNA were purchased from IDT.

**CEL-Seq2 and data analysis**. Pax7-CreERT2;Rosa26-YFP mice[39,40] were treated with tamoxifen to induce yellow fluorescent protein (YFP) in a Pax7-expressing SC population. Tamoxifen dissolved in corn oil (5 µl/g of 20 mg/ml) was injected intraperitoneally into mice daily for 3 days prior to the induction of muscle injury. Muscle injury was induced by the intramuscular injection of 100 µl CTX (10 µM; (Sigma-Aldrich, St. Louis, MO) into the gastrocnemius muscles of anaesthetized mice using a 29-gauge 1/2 insulin syringe. Regenerating muscles were isolated 1 day after CTX injection. Mononuclear cells from uninjured or injured gastrocnemius muscles (quiescent SCs and activated SCs) were prepared using 0.2% collagenase type II (Worthington Biochemical, NJ, USA) as previously described[41,42]. The YFP-positive SC population was sorted using a fluorescence-activated cell sorting (FACS) Aria II flow cytometer (BD Immunocytometry Systems, CA, USA). Debris and dead cells were excluded by forward scatter, side scatter, and PI gating. Data were collected using the FACS Diva software (BD Biosciences, CA, USA). Individual MFs were isolated from the EDL muscle as described previously[42]. Transcriptome analysis of quiescent SCs, activated SCs, and MFs were performed as previously described[17]. The libraries were sequenced using Illumina MiSeq and the reads were mapped to the mouse reference genome (mm10) by the Bowtie 2 software (version 2.2.6)[43]. The 3′-untranslated regions of the histone variants were determined by visualizing the alignments using the Integrative Genomics Viewer. Gene counting was performed using HTSeq[44] and the counts for a gene with the same UMI were counted as one (UMI counts). The Z-scores for each SC marker gene[45] were calculated with the UMI counts normalized using the DESeq2 software[46]. In addition, Pearson correlation coefficients were calculated with the common logarithm of the UMI counts for those genes with non-zero counts in at least one sample.

**Muscle regeneration studies**. Muscle regeneration studies were performed as previously reported, except for the injection of CTX into the TA muscle[47].

Immunohistochemistry in Fig. 1b was performed as described in ref. [48], except that ice-cold acetone was used for fixation instead of paraformaldehyde. Primary antibodies used were rat anti-laminin-2 alpha 4H8-2 (1:200) (L0663, Sigma-Aldrich) and mouse anti-dMHC (RNMy2/9D2) (1:50) (Leica Biosystems, Germany). Secondary antibodies were Alexa Flour 488-labeled goat anti-rat (1:800) (Thermo Fisher Scientific, Yokohama, Japan) and CF 568-labeled goat anti-mouse (1:800) (Biotium, Fremont, CA). Samples were viewed on a Keyence BZ-X700 microscope (Keyence, Osaka, Japan), following mounting with ProLong Diamond Antifade Mountant with 4,6-diamidino-2-phenylindole (DAPI; Thermo Fisher Scientific).

For the immunohistochemistry shown in Supplementary Fig. 4, at 5 days after CTX injection, three cryosections of the TA muscle were prepared from each mouse. Antibodies used were: rabbit anti-Pax7 (PA1-117, Thermo Fisher Scientific, 1:200), rat anti-MyoD[49] (1:200), rat anti-laminin-2 alpha 4H8-2 (L0663, Sigma-Aldrich, 1:200), and rabbit anti-laminin antibodies (L-9393, Sigma-Aldrich, 1:500). Images were visualized using a fluorescence microscope (BZ-9000; Keyence, ×40 objective). For each section, 4–9 high-power fields were combined. For quantification of Pax7- and MyoD-positive cells, threshold was determined using cytoplasmic signals of non-specific background, DAPI-negative areas were removed, and then areas with specific signals over threshold were counted. Similarly, MFs were defined as areas surrounded by laminin staining, and then the

number was counted. Image processing and cell counting were performed using the BZ-X analyzer software.

For cross-sectional area measurements: Frozen sections of mouse TA muscles were stained with hematoxylin and eosin. Images were obtained with a BZ-X700 microscope at 20× objective magnification. MF cross-section areas (µm²) were measured using the BZ-X analyzer (Keyence, Osaka, Japan).

**Cells**. C2C12 cells were grown in Dulbecco's modified Eagle's medium (DMEM) supplemented with 20% fetal bovine serum. Undifferentiated cells were harvested at 60–70% confluence. Differentiated cells were transferred to DMEM containing 2% horse serum upon reaching confluence. NIH3T3 cells were infected with retrovirus-expressing MyoD as described previously[10]. These cells were purchased from the American Type Culture Collection (ATCC, VA, USA).

**Generation of GFP-fused histone H3 variant cell lines**. *H3f3a* and *H3mm7* cDNAs were used for the expression of H3.3 and H3mm7 as previously reported[14]. H3.3 S57A and H3.3 T32I cDNAs were generated by the site-directed mutagenesis method for the *H3f3a* cDNA template. cDNAs were ligated into the Bidirectional Tet Expression Vector pT2A-TRETIBI (modified Clontech Tet-On system), which contains TolII transposon elements and an EGFP cDNA located upstream of the cDNA sequence (GFP-H3: N-terminus)[14]. pT2A-TRETIBI/EGFP-H3 variant transfection was performed using Lipofectamine 2000 (Life Technologies, Carlsbad, CA). NIH3T3 cells at 50% confluence were transfected with an expression vector (2 µg plasmid DNA per well in a 6-well plate), pCAGGS-TP encoding transposase (provided by Dr. Kawakami), and pT2A-CAG-rtTA2S-M2 and incubated for 24 h. To create cell lines with the stable expression of GFP-H3 variants, the transfected cells were cultured for 14–21 days in the presence of 1 µg/ml doxycycline and 1 mg/ml G418. GFP-positive cells were selected by FACS.

**Generation of H3mm7 KO C2C12 cell lines**. H3mm7 genomic loci were disrupted using the CRISPR/Cas9 system. Sequences of the form N20-NGG in the assigned region were first extracted using the Bowtie software (version 0.12.8). Then unique sequences, such that variations with less than three mismatches do not exist outside the considered region, were selected as candidates for gRNAs[50]. A gRNA sequence targeting H3mm7 (Chromosome 13: 117,358,212-117,358,622; GRCm38) was cloned into pSpCas9 (BB)-2A-GFP (PX458, Addgene). The plasmid was transfected into C2C12 sells using Lipofectamine 2000 Transfection Reagent (Thermo Scientific), in accordance with the manufacturer's instructions. To delete the H3mm7 gene locus completely, we performed transfection in three stages with three pairs of gRNAs. At 24 h after the transfection of the first pair, GFP-positive cells were isolated by cell sorting. The second pair of gRNAs was then transfected in these cells. Finally, at 24 h after the transfection of the third pair, GFP-positive cells were sorted into 96-well plates to establish single-cell clones that were subjected to genotyping by PCR. $H3mm7^{-/-}$ (homo deletion of H3mm7 loci) clones were obtained by detection using Sanger sequencing and amplicon sequencing. The gRNA and primers used in this study are listed in Supplementary Table 4. For rescue experiments, the pT2A-TRETIBI/H3mm7-EGFP or H3.3-EGFP cDNA plasmids (C-terminus) were stably transfected into $H3mm7^{-/-}$ C2C12 cell lines (it was confirmed that both C- and N-terminus tags of histone H3 produced comparable effects in C2C12[10,14]). GFP-positive cells were selected as rescue clones by FACS.

**FRAP**. FRAP was performed as described[51] using a confocal microscope (FV-1000; Olympus) with a 60× PlanApoN Oil SC NA = 1.4 lens. A confocal image of a field containing 2–5 nuclei was collected (800 × 800 pixels, zoom 1.2, scan speed 2 µs/pixel, pinhole 800 µm, 4 line averaging, BA505 emission filter, and 0.1% transmission of 488-nm Ar laser), one half of each nucleus was bleached using 100% transmission of a 488-nm laser, and images were collected using the original setting at 5 min intervals.

**Immunocytochemistry**. Cells were plated in 24-well plates (ibidi USA Inc., WI, USA), washed twice with phosphate-buffered saline (PBS), fixed with 1% paraformaldehyde in PBS for 10 min, permeabilized with 0.5% Triton X-100 in PBS, and washed twice with PBS. A 15 min incubation with Blocking One (Nacalai Tesque Inc.) was followed by a 2 h incubation with mouse anti-MHC (Calbiochem, 1:100; Fig. 1d) diluted with 10% Blocking One in PBS at room temperature. The wells were then washed three times with PBS and incubated for 30 min at room temperature with CF568-labeled goat anti-mouse (1:1000; Biotium Inc.) and bisbenzimide H33342 fluorochrome trihydrochloride (Hoechst) (1:5000; Nacalai Tesque Inc.) diluted with 10% Blocking One in PBS. Wells were washed three times in PBS and mounted in ibidi mounting medium (ibidi USA, Inc.). Images were visualized using a fluorescence microscope (BZ-9000; Keyence). The fusion index was evaluated using the ImageJ software.

**Immunoblotting**. Cells were washed twice with PBS, centrifuged, and then resuspended in 2× sodium dodecyl sulfate (SDS) sample buffer. The samples were separated by SDS-polyacrylamide gel electrophoresis (PAGE) and transferred to a polyvinylidene fluoride membrane with a Trans-Blot Turbo Transfer System

(Bio-Rad Laboratories, Hercules, CA). The membrane was blocked for 1 h in 5% (w/v) skim milk in Tris-buffered saline containing 0.05% (v/v) Tween 20 (TBST), then incubated with primary antibodies in Hikari Solution A (Nacalai Tesque Inc.) followed by incubation with horseradish peroxidase-labeled secondary antibodies and detection using Chemi-Lumi One Ultra (Nacalai Tesque Inc.). The primary antibodies used in Supplementary Fig. 7 included rabbit anti-Hsp90 (H-114, Santa Cruz Biotechnology, 1:1000) and anti-H3-C (1G1, hybridoma supernatant, 1:1000)[52]. Secondary antibodies were horseradish peroxidase-conjugated anti-rabbit and anti-rat IgG antibodies (NA934 and NA935, GE Healthcare, 1:5000). Cells were fractionated into soluble and chromatin fractions[51] and the total amount of H3 in these fractions was determined. Cells were washed twice with PBS and then lysed in ice-cold physiological buffer (100 mM $CH_3COOK$, 30 mM KCl, 10 mM $Na_2HPO_4$, 1 mM DTT, 1 mM $MgCl_2$, 1 mM ATP, 0.1% Triton X-100, and protease inhibitor cocktail). Lysates were incubated for 10 min on ice and then centrifuged at $1000 \times g$ for 5 min. The supernatants including soluble fractions were pelleted to be concentrated by trichloroacetic acid precipitation. The pellets and chromatin were resuspended in 2× SDS sample buffer. Lamin B1 (12987-1-AP, Proteintech, 1:1000) antibody was used as nuclear loading control. Uncropped images are given in Supplementary Fig. 24.

**Quantitative RT-PCR**. Total RNA was isolated and reversed-transcribed with PrimeScript Reverse Transcriptase (Takara Bio Inc.) and an oligo dT primer according to the manufacturer's instructions. qPCR was performed using Thunderbird qPCR Mix (Toyobo Co., Ltd.). Primers are listed in Supplementary Table 5. qPCR data were normalized to *Gapdh* expression levels and presented as the mean ± standard deviation of three independent experiments.

**Variant-specific qRT-PCR**. Allele-specific real-time PCR was employed to detect the expression of H3mm7[53]. SCs were expanded from Pax7-positive SCs, most of which were activated[54]. Real-time PCR amplification and detection were performed in a Pico Real-Time PCR machine under standard thermocycling conditions: 30 s at 95 °C, 45 cycles of 5 s at 95 °C, 30 s at 55 °C, and 30 s at 72 °C. The expression level of H3mm7 was evaluated as the ratio relative to the level of *H3f3a*. The primers used are listed in Supplementary Table 6.

**Chromatin immunoprecipitation**. ChIP assays were performed as described previously with some modifications[9]. Briefly, cultured cells were cross-linked in 0.5% formaldehyde and suspended in ChIP buffer (5 mM PIPES, 200 mM KCl, 1 mM $CaCl_2$, 1.5 mM $MgCl_2$, 5% sucrose, 0.5% NP-40, and protease inhibitor cocktail; Nacalai Tesque Inc.). The samples were incubated for 15 min on ice, sonicated for 5 s three times, and digested with micrococcal nuclease (1 μl; New England Biolabs, Ipswich, MA) and Ribonuclease A (1 μl; Nacalai Tesque Inc., Kyoto, Japan) at 37 °C for 40 min. The digested samples were centrifuged at $15,000 \times g$ for 10 min. Supernatant containing 4–8 μg DNA was incubated with a rat monoclonal antibody against GFP (Bio Academia Co. Ltd., 1A5, 2 μg) and mouse monoclonal antibodies against H3K4me3 (2 μg)[55] or H3K27me3 (2 μg)[56] pre-bound to magnetic beads at 4 °C overnight with rotation. The immune complexes were eluted from the beads using 1% SDS in TE, followed by washing with ChIP buffer and TE buffer (both twice). Cross-links were reversed, and DNA was purified by a Qiaquick PCR purification Kit (Qiagen, Valencia, CA).

**ChIP-seq**. ChIP libraries of GFP, H3K27me3, and H3K4me3 were prepared using the NEBNext Ultra DNA Library Prep Kit for Illumina (New England Biolabs) with biological duplicates. The libraries were sequenced on Illumina HiSeq 1500 sequencers. The data were aligned to mouse genome (GRCm38) using HISAT2 (version 2.0.4), and the uniquely mapped reads were retained for the following analysis. For the visualization of ChIP-seq signals on the Integrative Genomics Viewer, read counts of 100 bp bins in the genome were normalized as reads per million (RPM; multiplying a scaling factor of $10^6$/total number of reads). The RPM counts of each bin were then smoothed by averaging each center ±5 adjacent bins (smoothing with 1000 bp window).

**RNA-seq**. Total RNA at the growth and differentiated (i.e., post-differentiation) states for C2C12 cells and NIH3T3 cells were obtained as described above. Library preparation and sequence analysis were performed according to the manufacturer's protocol (NEBNext Ultra Directional RNA Library Prep Kit for Illumina). In the analysis of $H3mm7^{-/-}$ C2C12 cells using DESeq2 software, the interaction effect was tested using a likelihood ratio test with the full model: Time (Growth, 24, 48 and 72 h) + Condition ($H3mm7^{-/-}$ and $H3mm7^{+/+}$ cells) + Time × Condition (interaction term), vs. the reduced model: the full model without the interaction. In response to the result of the test (no genes had interactions), the DEGs from the reduced model were employed in the subsequent analysis. The contrasts of the Condition term (log2FC of $H3mm7^{-/-}$ vs. $H3mm7^{+/+}$) of 3505 genes met the criteria: Benjamini and Hochberg corrected Wald $p < 0.1$ and baseMean >100 (the average expression level for all conditions). The DEGs in the RNA-seq data of NIH3T3 cells, which have four time points (UI, 0, 24, and 48 h) and three conditions (H3.3+, H3mm7+, and WT), were extracted by fitting the model without interaction as performed in C2C12 cells. The preranked GSEA[57] with the log2FC values of DEGs was performed using the fgsea package[58] and visualized

using the DOSE package[59] in R. The regularized log-transformed (rlog function in DESeq2) read counts of genes were utilized as the gene expression level in the following analysis. *K*-means clustering, kmeans function in R with the options: $k = 6$, algorithm = "Lloyd", nstart = 200, iter.max = 1000, was performed against the regularized log counts of the extracted DEGs. Linear discriminant analysis was done using the lda function in the MASS package of R with the given class label of the clusters.

**ATAC-seq**. For ATAC sequencing, 50,000 sorted cells were centrifuged at $800 \times g$ for 5 min, followed by a wash using 500 μl of cold PBS and centrifugation at $800 \times g$ for 5 min. Cells were lysed using cold lysis buffer (10 mM Tris-HCl, pH 7.4, 10 mM NaCl, 3 mM $MgCl_2$, and 0.1% IGEPAL CA-630). Immediately after lysis, nuclei were spun at $500 \times g$ for 10 min using a refrigerated centrifuge. Next, the pellet was resuspended in the transposase reaction mix (10 μl 2× TAPS + DMF buffer, 1.5 μl transposase (in house)[60] and 8.5 μl nuclease-free water). The transposition reaction was carried out for 30 min at 37 °C and immediately placed on ice. Directly afterwards, the sample was purified using a Qiagen MinElute Kit. Following purification, the library fragments were amplified using custom Nextera PCR primers 1 and 2 for a total of 15 cycles. Following PCR amplification, the libraries were purified using a Qiagen MinElute Kit. Paired-end sequencing was performed with 100 cycles on an Illumina HiSeq 1500 platform. The differentially accessible genes were extracted using DESeq2. We employed an index of gene accessibility as the ATAC-seq read counts in a promoter region (2 kb upstream from TSS). The fitted model is: Time (UI, 0, 24, and 48 h) + Clone (H3mm7+, H3.3+, and WT cells), with the option: fitType = "local".

**Purification of nucleosomes with mouse histone variants**. All recombinant histones (H2A, H2B, H4, H3.3, H3mm7, and H3.3 mutants) were bacterially produced and purified by Ni-NTA agarose chromatography followed by MonoS chromatography (GE Healthcare) as previously described[61,62]. To form histone complexes (H2A–H2B dimer and H3–H4 tetramer), H2A and H2B or H3 and H4 were mixed at an equimolecular ratio in 20 mM Tris-HCl (pH 8.0) buffer, containing 7 M guanidine hydrochloride and 20 mM 2-mercaptoethanol. After the samples were dialyzed against a refolding buffer (10 mM Tris-HCl (pH 7.5), 1 mM EDTA, 2 M NaCl, and 2 mM 2-mercaptoethanol), histone complexes were purified by chromatography on Superdex 200 gel filtration column (GE Healthcare), as previously described[63]. The nucleosomes were reconstituted with the H2A–H2B dimer, the H3–H4 tetramer, and the 146 base-pair DNA derived from human alpha-satellite DNA by the salt-dialysis method[63]. The reconstituted nucleosomes were purified using Prep Cell apparatus (Bio-Rad)[64].

**Crystal structures for the mouse H3.3 and H3mm7 nucleosomes**. Purified nucleosomes containing H3.3 or H3mm7 were crystallized by the hanging-drop vapor diffusion method[62]. For X-ray diffraction experiments, nucleosome crystals were soaked in a reservoir buffer (20 mM potassium cacodylate (pH 6.0), 45–60 mM $MnCl_2$, 35–45 mM KCl) supplemented with 30% 2-methyl-2,4-pentanediol and 2–3% trehalose. The crystals were flash-cooled in liquid nitrogen. Diffraction data of the nucleosomes were collected at BL41XU beamline station in SPring-8 (Harima, Japan) and at BL1A, BL5A, and BL17A beamline stations in Photon Factory (Tsukuba, Japan). The diffraction data were analyzed by HKL2000 and CCP4[65,66]. The structures were solved by the molecular replacement method with the *Phaser* program using the structure of the human H3.3 nucleosome (PDB ID: 3AV2) as the search model[64,67]. The structural refinement was performed by the PHENIX and COOT software[68,69]. The H3.3 and H3mm7 nucleosome structures were determined at 2.87 Å and 3.45 Å resolutions, respectively. The structural data have been deposited in the RCSB Protein Data Bank [PDB ID: 5XM0 (for H3.3 nucleosome) and 5XM1 (for H3mm7 nucleosome)]. All structure figures were generated with the PyMOL software (http://www.pymol.org). The detailed list of the data collection and refinement statistics are shown in Supplementary Table 7.

**Thermal stability assay**. For the thermal stability assay of the nucleosomes, purified nucleosomes were mixed with 5× SYPRO Orange (Sigma-Aldrich) in 10 mM Tris-HCl buffer (pH 7.5) containing 100 mM NaCl and 1 mM dithiothreitol, and the experiments were performed as described previously[27]. The intensities of fluorescence signals emitted from SYPRO Orange bound to denatured histones were normalized relative to the signal at 95 °C.

**Salt-resistance assay**. For salt-resistance assays of the nucleosomes[62], purified nucleosomes were incubated at 55 °C in 10 mM Tris-HCl buffer (pH 7.5), containing 1 mM dithiothreitol and NaCl (0, 500, 600, or 700 mM), for 1 h. The samples were then analyzed by non-denaturing PAGE stained by ethidium bromide.

**Code availability**. The codes used for our deep-sequencing data analysis are available at https://github.com/kazumits/DataAnalysisForH3mm7.

**Data availability**. The next-generation sequencing data are deposited in GEO [GEO: GSE104389] and in the DNA Data Bank of Japan [DDBJ: DRA005975,

DRA006453], and the structures of the H3.3 and H3mm7 nucleosomes are archived at Protein Data Bank [PDB: 5XM0, 5XM1].

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

## Acknowledgements

We thank the Advanced Computational Scientific Program of the Research Institute for Information Technology, Kyushu University and the National Institute of Genetics (NIG) for providing high-performance computing resources. We also thank the Joint Usage and Joint Research Programs, the Institute of Advanced Medical Sciences, Tokushima University and beamline scientists at the Photon Factory for their assistance with data collection from the SPring-8 BL41XU beamline and the BL-1A, BL-5A, and BL-17A beamlines. The synchrotron radiation experiments were performed with approval of the Japan Synchrotron Radiation Research Institute (JASRI) (Proposal 2013A1036 and 2014A1042) and the Photon Factory Program Advisory Committee (Proposal 2012G569, and 2014G556). This work was supported by JST CREST [JPMJCR16G1]; MEXT/JSPS KAKENHI [JP25116010, JP25132709, JP25118518, JP26290064, JP17H03608, and JP17K19356 to Y.O.; JP16H01219 and JP15K18457 to A.H.; JP16K18479, JP16H01577, and JP16H01550 to K.M.; JP25116002 and JP17H01408 to H.Ku.; and JP25116005 to H.Ki.] and is also partially supported by the Platform Project for Supporting Drug Discovery and Life Science Research [to H.Ku.] and the Practical Research Project for Allergic Diseases and Immunology (Research on Allergic Diseases and Immunology) [JP17ek0410044 to Y.Oh.] from the Japan Agency for Medical Research and Development (AMED). H.Ku. was also supported by the Waseda Research Institute for Science and Engineering and by programs of Waseda University. H.T. was supported by a Research Fellowship of the Japan Society for the Promotion of Science for Young Scientists.

## Author contributions

A.H., K.M. and Y.Oh. designed the experiments. A.H. and Y.Se. performed deep-sequencing analyses. K.M. and J.N. performed computational analyses. H.T., Y.X. and H.Ku. performed structural and biochemical analyses. Y.On., K.Y., Y.K., T.I. and S.O. analyzed knockout mice. Y.Sa. and H.Ki. performed FRAP experiments. T.T. generated knockout mice. A.H., K.M., T.K., H.Ki., H.Ku. and Y.Oh. wrote the manuscript. All authors read and approved the final manuscript.

## Additional information

**Competing interests:** The authors declare no competing interests.

