## [Peer Review File · Nature Communications]

Reviewers' comments:

Reviewer #1 (Remarks to the Author):

In this study, Harada and colleagues investigate the functional importance of a newly identified subvariant of histone H3, a so called H3mm7, which is preferentially expressed in skeletal muscle satellite cells. The authors generated H3mm7 knockout mice and H3mm7 knock out cell lines and used muscle regeneration and cell biology experiments, including PCR and ChIP, to demonstrate that H3mm7 facilitates target/myogenic gene transcription in vivo and in vitro. The authors detail a new crystal structure of the H3mm7-containing nucleosome and hypothesize that the replacement of S57 in H3.3 with A57 in H3mm7 results in a less stable nucleosome.

This work sheds light on the critical role of tissue/cell specific histone variants. It will be a valuable contribution to our understanding of lineage specification as well as to the area of chromatin biology in general. A combination of techniques used in this study is commendable, and the experimental design and data analysis are of excellent quality. I have only a few specific points:

1. The second part, describing the structure of the H3mm7-containing nucleosome, reads as a somewhat detached study. A better link between biology experiments and the nucleosomal structure and biochemical data would greatly strengthen this manuscript.
2. The H3.3 T32I-containing nucleosome seems to disassemble faster than the H3.3-containing nucleosome, shown in Fig. 4c. It would be interesting to assess the role of T32 in H3.3 vs I32 in H3mm7. Could T32 be phosphorylated? To what extent does the hydrophobic character of I32 disrupt association of H3 tail with the nucleosomal core, decreasing the nucleosome stability? The reciprocal mutant, H3mm7 I32T, should be tested here.
3. Is it feasible to examine the T32I and S57A mutants in vivo? If yes, this would well connect biological and structural parts.

R40 adopts the same conformation and is positioned similarly regardless of the formation of the hydrogen bond with S57. Some more convincing data are needed to make conclusions about the significance of the hydrogen bond between R40 and S57.

Reviewer #2 (Remarks to the Author):

The authors' research group previously identified a histone H3.3 sub-variant, H3mm7. Here, they tried to investigate the functional roles of H3mm7 during cell differentiation. They claimed that H3mm7 is required for normal skeletal muscle regeneration based on phenotype analysis using H3mm7 knockout mice. Although the crystal structure analysis demonstrate H3mm7-specific A57 residue cannot form a hydrogen bond with the R40 residue of the H4 molecule, the mechanism underlining muscle regeneration deficiency in H3mm7^{-/-} mice remains to be revealed.

Major points:

1. The authors claimed that H3mm7 is specifically expressed in mouse skeletal muscle tissue (Page 5, lines 93-94). However, when I double checked the reference 14, no convincing data demonstrate such tissue specificity of H3mm7 gene expression.
2. In Figure 1A, their data showed that H3mm7 was highly expressed in quiescent satellite cells (SCs) and dramatically decreased after SC activation. In myofibers, expression level of H3mm7 was very low. The data suggest H3mm7 expression is downregulated during myogenic cell differentiation. Thus, how do they think about the observation that deletion of H3mm7 in C2C12 cells delayed cell differentiation (Figure 1D, 1E).
3. H3mm7 knockout mice exhibit delayed muscle regeneration based on regenerating fiber size and eMHC gene expression. The further experiments should be performed to clarify which cellular

event(s) is/are affected by H3mm7 deletion during muscle regeneration: satellite cell activation? Proliferation? Differentiation? Or self-renewal?

4. Next, they tried to understand the mechanism underlining delayed muscle regeneration by RNA-seq. Why do they perform such experiments with C2C12 cells but rather do RNA-Seq using satellite cells from H3mm7^{-/-} mice?

5. The authors claimed that H3mm7 might facilitate gene expression by incorporating into nucleosome. After deletion of H3mm7, one should expect the decreased expression level of those H3mm7-regulated genes. How do they think about the RNA-seq data that almost the same number of genes was upregulated in H3mm7^{-/-} C2C12 cells (Figure 2B)? Similarly, why did they observe almost the same number of genes was downregulated in NIH3T3 cells with ectopic expression of H3mm7 (Figure 3B)?

Reviewer #3 (Remarks to the Author):

In their manuscript "Histone H3.3 sub-variant H3mm7 is required for normal skeletal muscle regeneration" Harada et al. investigate the specific role of the recently identified histone variant H3mm7 on muscle regeneration in mouse. They find that H3mm7 is specifically expressed in satellite cells and is necessary for proper muscle regeneration after injury in vivo and also for muscle differentiation in vitro. Their data suggest that presence/absence of H3mm7 may specifically affect the expression of genes upregulated during muscle differentiation. This could be due to the fact that nucleosomes containing H3mm7 are less stable than H3.3 containing nucleosomes and would therefore favour transcription of a subset of genes.

I find the paper original and interesting in the context of histone variants and their specific roles in chromatin activity. The authors use several approaches offering different angles to study functional differences between H3mm7 and H3.3, which differ from each other only by two amino acids. It is rather clear that H3mm7 depletion impair muscle regeneration in vivo and differentiation of C2C12 cells in vitro. I think also that the data support well a model where nucleosomes containing H3mm7 are less stable than H3.3 containing nucleosomes. I found however more difficult to appreciate the specific role of H3mm7 on gene expression during differentiation. I feel that more data and robust statistical analysis should be included in the manuscript to firmly conclude that H3mm7 expression modulates the transcriptional activity of genes that are differentially expressed across differentiation and to support a role distinct from H3.3.

Major points

1. Supplementary Fig4: there is no statistical analysis. I think this should be done considering the variation observed for some time points, in particular in the H3mm7^{-/-} cells. There is also a difference between the number of "measurements" between the different conditions: for example, 14 points at 48h for H3mm7^{-/-} vs 7 points for H3mm7^{+/+} at the same time point. The standard deviation bars are also a bit surprising: how can it be so small in the case of H3mm7^{-/-} cells at 48h with such a distribution of values?

2. Supplementary Fig5: the western blot is not convincing. A picture showing a lower exposure for H3 should be shown (where the level of H3 for each lane could be clearly seen) and a clean immunoblot should be shown for H2B (or no blot at all and remove the statement on H2B levels in the main text since it does not add anything to the main conclusion). Since the levels of Hsp90 detected in the 4 lanes are different, it is important to show clear immunoblots for H3 and quantification of the signals (related to Hsp90) may be useful here.

3. Page 7, lines 146-147: More genes would be necessary to support the conclusion of the authors. Indeed, one gene only (Ckm) is fully supporting a function unique to H3mm7 in the regulation of genes associated to skeletal muscle differentiation. The other gene (Des) however does not support this conclusion.

4. Page 10, lines 200-201: The data supporting the conclusion are not strong enough. Indeed, only two replicates were done for each cell type (and each time point) and the two replicates for the H3mm7+/+ show variations that make the interpretation of the results difficult. This is particularly the case for the house keeping genes (showing no variations across the differentiation process) that were detected as DEGs by DESeq2. How to be sure that H3mm7 depletion has really no effect on the expression of these genes and only modifies the expression of genes associated with differentiation? To support the idea that H3mm7 only regulates the expression of genes affected by the differentiation process, a third replicate should be included in the analyses (at least for H3mm7+/+ cells) to verify that housekeeping genes expression is not altered in absence of H3mm7 and that the results regarding the other genes are not due to variations between replicates.

5. Supplementary Fig11 : The calculation of a difference between mean values is a weak method to support the fact that H3mm7 overexpression has more effect on gene expression than H3.3. The data showed in the right panel should at least contain standard deviations. What test was used to determine the p-values in that particular case?

6. Page 10, lines 216-218 (and Supplemental Fig11): To show that "the expression of H3mm7 led to more rate change [...] compared with that caused by H3.3" the authors should compare the ratios H3mm7/control and H3.3/control (in terms of gene expression, for each gene) and verify whether or not they are globally, statistically different. The data are not strong enough as they are presented to support the conclusion.

7. Page 11, lines 222-223 and Supplementary Fig12: the qPCR results are actually not fully supporting the RNA-seq data. Indeed, in the case of Ckm, H3.3 and H3mm7 seem to have the same effect on gene expression. In the case of Myh3, it is even more surprising since at 48h, H3mm7 expression has no significant effect on gene expression while H3.3 has. To which cluster does Myh3 belong (my guess is UA)? Regarding the expression of this gene, overexpression of H3mm7 and H3.3 appear to have opposite effects, which is not the case in any of the clusters shown in Fig3B. Again, more genes here would be necessary to get a better picture of the effect of H3mm7 on gene expression compared to H3.3 in NIH3T3 cells.

8. Page 11, lines 238-241: Another obvious conclusion from this part is that H3.3 and H3mm7 are deposited overall at the same places within the genome. This goes in the direction of overlapping functions between the 2 variants.

9. Page 12, lines 252-253 and Fig3C: There are no statistics or numbers to support that "the chromatin accessibility and expression of the genes belonging to cluster UA were enhanced with H3mm7 compared with H3.3". Moreover, the figure only shows the difference in gene expression between H3mm7 expressing cells and control cell without overexpression of H3.3. Fig3B addresses this point so if the authors conclusion is also based on panel B, they should refer to it as well.

10. Page 12 lines 260-262: There is so far no strong evidences that H3mm7 has (i) a specific role in the expression of genes related to myogenic differentiation and (ii) compared to H3.3.

11. Page 13, line 284 – page 24, line 287 and Fig4E: in which cell type was the FRAP experiment done? More cells would have made the conclusion stronger. The authors are mentioning a faster turnover of H3mm7 and H3.3S57A. However, looking at the panels showing the photobleaching (0 min) it appears that the bleaching is incomplete in the case of H3.3S57A and to a lower extent with H3mm7. This could be due to the presence of an insoluble pool that is not incorporated rather

than just rapid turnover (due to instability). This could be tested by doing a simple Triton TX-100 extraction to separate the soluble and insoluble fractions. A western blot showing the level of expression of the different exogenous proteins should also be done to verify that the presence of a soluble pool is not due to a high overexpression.

Minor points

In the RESULTS section:

1. It is difficult to understand how the CEL-Seq2 experiment was conducted (page 5, lines 98-104). First the author say that "global gene expression [...] was performed at different stages of myogenesis using quiescent or activated satellite cells and myofibers isolated from the extensor digitorum longus (EDL) muscle". Then they refer to a muscle injury induced by injection of CTX into the gastrocnemius muscle in a certain type of mice to explain the origin of the cells they mentioned earlier. The procedure used should be put before the results referring to Fig1A.
2. Page 5, line 105: Pax3 is not expressed in activated satellite cells according to Fig1A so H3mm7 should not be mentioned as synchronously expressed with Pax3 in these cells (but with Pax7, yes).
3. Which satellite cells (activated or inactivated) were used to assess the cell-specificity of H3mm7 expression by RT qPCR in Supplementary Fig2? It is not indicated in the text, figure legend or methods.
4. In Supplementary Fig2, why is H3mm7 expression normalized to H3f3a and not Gapdh as for H3f3a and H3f3b? H3mm7 expression normalized to Gapdh should be shown as well.
5. Supplementary Fig2: The legend should be re-written.
6. Page 5, lines 108 – 110: Why the fact that H3f3a also shows a decreased expression in myofibers participates to the conclusion "H3mm7 functions in earlier stages of skeletal muscle regeneration".
7. The "H3mm7 knockout mouse line" is not documented at all. The authors should at least include a figure showing that these mice lack the H3mm7 gene.
8. Fig1D: the Hoechst staining is almost impossible to see in the upper panels. It is thus very difficult to visualize the undifferentiated cells, especially among the H3mm7^{-/-} cells (top-right). Showing cells with higher magnification would strengthen the conclusion.
9. Page 6, lines 113-115: "H3mm7 knockout mice are viable" is a weak argument for stipulating that "H3mm7 is not necessary during development". If the authors mean that these mice are indistinguishable from control mice, they should mention it instead. Indeed, "being viable" does not mean to be "normal" and these mice could suffer from various malformations, which does not seem to be the case.
10. Page 6, line 117: a reference to Fig.1B should be put after "[...] CTX injection".
11. Page 6, line 118: a reference could be added at the end of the sentence (regarding expression of dMHC during muscle regeneration).
12. Fig1B: the red signal is weak and could be brighter so it would be easier to see.
13. Page 6, line 128: a reference to Fig1D (lower panels) could be added at the end of the sentence. I would also specify "ectopic expression of ..." instead of just using "H3mm7-GFP".

14. Page 6, line 131: the authors do not discuss at all why ectopic expression of H3.3 could worsen the phenotype observed in H3mm7^{-/-} cells. This is an interesting observation that should be discussed.
15. Page 6, line 132: What is "the H3-C antibody"? Even though the reference (44) is indicated in the methods, it should be mentioned here as well.
16. Page 6, lines 131-133: The cell types that are compared should be mentioned.
17. Page 7, lines 137-138: Eef1a1 is not a muscle marker, as mentioned later (used as a control, housekeeping gene). Thus only 2 representative skeletal muscles markers are used.
18. Fig1E: What are the orange and blue bars? It is not indicated in the figure or in the figure legend. Also, the reference bar corresponding to expression at 48h in H3mm7^{+/+} cells (=1) should also show a standard deviation. To do so, each value of the reference sample should be divided by the mean value for this sample. With the resulting values, one can get a standard deviation centred on 1.
19. Page 7, lines 137-138: The authors write that "the mRNA level of Ckm was decreased in H3mm7^{-/-} cells at 48 h after the differentiation stimulus". This sentence is confusing since it may indicate a decrease of Ckm expression upon induction of differentiation, and that is not the case. I think the authors should re-write this sentence.
20. Page 7, lines 142-147: The authors claim that "H3.3 and H3mm7 both rescued Des expression", relying on p values of 0.064 and 0.032 respectively, even though Fig1E shows no statistical differences in case of " H3mm7^{-/-} +H3mm7". However, they write just after that "levels of the control [...] gene Eef1a1 were not significantly changed in H3mm7^{+/+}, H3mm7^{-/-} or the rescued clone" while Fig1E shows a statistically significant increase of expression in the "H3mm7^{-/-} + H3.3" experiment. This part should be re-written taking into account the statistical significance, or not, of the differences.
21. Supplementary Fig6: I am not familiar with DESeq2 and I cannot evaluate the method. However, I have the feeling that a Y-axis legend is missing to better understand the cartoon. Is it gene expression level?
22. Page 8, lines 159-162: "Loss of gene expression [...]": Why is this sentence here? Is it to show that the method used by the authors (DESeq2) is working in other contexts? If so, the author should at least say it.
23. Supplementary Fig7 : What is MB? MT? cond? si-mH2A? Is mH2A.1.2i the same thing as si-mH2A? Why are the points corresponding to the same time point not all aligned (y-axis). Is it normal?
24. Supplementary Fig8 : What is cond? Same as for Supplemental Fig7.
25. Page 8, line 167: Why "even"?
26. Page 9, lines 180-181: The way the conclusion is written tends to suggest that H3mm7 is not specifically involved in the expression of genes related to myogenic differentiation. Is it what the authors mean?
27. Fig2B: the legend on the right side of the graphs (under cluster) shows SA and SB clusters, while this is not shown in any graph. This should be removed.

28. Page 9, line 188 : "Flat staying patterns"? What is this?
29. Supplementary Fig9 : What is the y-axis and where are the values? Where is the legend for the symbols used in the graphs? What is G? Why using D24-72 in this figure while using 24-72h in other figures?
30. Page 9, lines 191-192: The conclusion is confusing. "The rate change of gene expression" between what and what? "Up- or down-regulation of gene expression" occurring when? The authors juggle the concepts of (i) gene regulation across differentiation and (ii) differentially expressed genes between H3mm7+/+ and H3mm7-/- cells. This makes the reading difficult.
31. Page 9, line 193 : "We next evaluated the effect of rate change...". The effect on what?
32. Fig2D: what are LD3, LD4 and LD5? This is not said anywhere.
33. Supplementary Fig10 : What chromosome is shown on the figure? The Y-axis has no label or legend.
34. Page 10, line 207: " [...] GFP-fused H3.3/H3mm7 NIH3T3 cells...". This should be better explained.
35. Page 11, lines 223-225: the way the conclusion is written emphasizes a role of H3.3 in gene regulation instead of minimizing it. The sentence should be re-written. And again, I am not convinced that the effect of H3mm7 expression in NIH3T3 cells differs globally from H3.3 overexpression, for the reasons mentioned above.
36. Page 11, lines 229-231: The link between this introductory sentence and the following part is not obvious.
37. Page 11, lines 231-233: In which cells type(s) were this experiments performed?
38. Supplementary Fig13 : There is no statistics supporting enrichment of H3.3 and H3mm7 in the mentioned regions.
39. Fig3C: What is "Weaker than H3.3" and why does it appear at two different places in the graph? Should one of them be "Greater than H3.3" (top-left maybe)? And what is "Enhanced"? What does "Enhanced (negative)" mean? The time point should be indicated in the figure legend.
40. Fig3D: Do Kif16b and Lmod2 show increased expression upon H3mm7 expression? It does not seem so. Is it normal for Gm20756 and Kif16b expression to show such a "flat" curve across differentiation while they belong to cluster UA?
41. Page 12, line 258: "Genes in SB (yellow) with [...] where the effect of H3.3 was stronger". Yellow where? Where are these data shown? Effect of H3.3 on what? I do not understand this sentence.
- In the METHODS section:
42. Page 20, lines 420-421: "[...] and an EGFP cDNA located upstream of the cDNA sequence [...]". The fused proteins should then be referred to as "EGFP-H3.x". However, in several places in the text and in figures, the authors use "H3.3-EGFP" or "H3mm7-EGFP", which would indicate that EGFP is fused to the C-terminal part of the histones (which personally, I would find more logical). The authors should correct this.
43. Page 20, lines 423-424: "[...] (2µg plasmid DNA per 6-well plate)". What does that mean? Is it

2 µg per well or something else?

44. Page 20, line 428: "fluorescence activating cell sorting" is already mentioned before as "FACS" (page 18, line 377). The meaning of FACS should be indicated earlier (page 18) and only the acronym used afterwards.

45. Page 21, line 461: the authors should indicate the incubation time for fixation.

46. Page 25, lines 539-540: "[...] which have three timepoints (UI, 0, 24 and 48 h) and ...". Isn't it four time points?

Erwan Delbarre
Ph.D.

Reviewer #4 (Remarks to the Author):

The manuscript "Histone H3.3 sub-variant H3mm7 is required for normal skeletal muscle regeneration" by Harada and others described how the structurally more unstable H3.3 variant H3mm7 is incorporated at the same promoters as H3.3, thereby creating more accessible chromatin state, resulting in a general change in expression rate. The authors used a true tour-de-force with a wide range of techniques to probe the function of H3mm7, ranging from generating a H3mm7 KO mouse and C2C12 cell line using CRISPR, immuno-staining, CEL-seq2, ATAC-seq, ChIP-seq, RNA-seq, FRAP, and crystallography.

KO of H3mm7 did not result in up- or down-regulation of specific genes, but rather a more general change in expression rate. This included genes involved in skeletal muscle differentiation. Furthermore, the authors determined that H3mm7 is expressed in earlier stages of skeletal muscle regeneration, most notably quiescent skeletal satellite cells. Indeed, a H3mm7 KO mouse displayed a subtle, but distinct phenotype, namely a delay in muscle regeneration.

Histone variants are thought to evolve by the birth-and-death paradigm. Previously, the authors discovered several H3.3 variants with limited phylogenetic spread, but potentially with functional consequences. Indeed, in this manuscript they delve into the functional implications of one of these novel H3.3 variants, H3mm7. The combination of the KO mouse, various sequencing techniques, and crystallography, a very nice correlation was drawn how subtle structural changes in a minor histone variant impacts global transcription by creating a more open chromatin structure, which in turn resulted in subtle phenotype.

Overall, we found this manuscript a pleasure to read and, it provides novel insight in how rare histone variants, through relatively subtle means, enhance specific outcomes, in this case, skeletal muscle cell differentiation.

There are only a few minor concerns that, in our view, do not preclude acceptance, but might improve readability.

Minor concerns:

1. On lines 131-3, the authors mention that no significant difference was found in total amount of H3. Is there a difference between soluble and chromatin-incorporated H3?
2. On line 165, the author's mention the "KO effect through time", but no reference is provided.
3. Typo on line 194: Fisher's linear (not liner) discriminant analysis.
4. Line 213, remove 'as' from 'as for C2C12 cells'.
5. Line 214, the authors mention "before infection (UI)". Might the authors have meant induction?

6. On line 339, the paragraph starts with a space, rather than a tab.

7. In Figure 1A, besides H3mm7, H3mm11 appears to have a very similar expression profile, just like the two H3f3a/b genes. Could H3mm11 function synergistically with H3mm7 to change the expression landscape during differentiation, based on tissue expression pattern and that H3mm7 is predominantly found at the promoter and H3mm11 predominantly found at TSS? Maybe the authors could speculate a little in the discussion, to make it a bit more fun for the lay reader?

-Yamini Dalal and Daniel Melters

NCI

Response to Reviewers' comments:

Our responses are given below in **black** and modifications made in the text are indicated in **red**.

Reviewers' comments:

Reviewer #1 (Remarks to the Author):

In this study, Harada and colleagues investigate the functional importance of a newly identified subvariant of histone H3, a so called H3mm7, which is preferentially expressed in skeletal muscle satellite cells. The authors generated H3mm7 knockout mice and H3mm7 knock out cell lines and used muscle regeneration and cell biology experiments, including PCR and ChIP, to demonstrate that H3mm7 facilitates target/myogenic gene transcription in vivo and in vitro. The authors detail a new crystal structure of the H3mm7-containing nucleosome and hypothesize that the replacement of S57 in H3.3 with A57 in H3mm7 results in a less stable nucleosome.

This work sheds light on the critical role of tissue/cell specific histone variants. It will be a valuable contribution to our understanding of lineage specification as well as to the area of chromatin biology in general. A combination of techniques used in this study is commendable, and the experimental design and data analysis are of excellent quality. I have only a few specific points:

We appreciate the reviewer's understanding of our work and are grateful for the comments, which greatly helped us strengthen our results and improve our presentation. We have performed additional experiments and data analysis to address your concerns.

1. The second part, describing the structure of the H3mm7-containing nucleosome, reads as a somewhat detached study. A better link between biology experiments and the nucleosomal structure and biochemical data would greatly strengthen this manuscript.

To address this comment, we further evaluated the regulatory function of the H3mm7 A57 residue in transcription. We overexpressed the H3.3S57A mutant in NIH3T3 cells and performed RNA-seq (cf. our response to Comment #3 for details). If the A57 residue of H3mm7 has a specific function, the H3.3S57A mutant may mimic the H3mm7-mediated transcription regulation. Interestingly, the data showed that exogenous production of H3.3S57A resulted in increased transcription levels of myogenic genes, including *Myog*, *Myh3* and *Tnnt2*, as presented in Supplementary Fig 23. Therefore, the H3mm7-specific A57 residue functions in up-regulation of transcription for these myogenic genes.

These results strengthen the biological relevance of the H3mm7 A57 residue, which contributes to the structure and stability of the nucleosome.

2. The H3.3 T32I-containing nucleosome seems to disassemble faster than the H3.3-containing nucleosome, shown in Fig. 4c. It would be interesting to assess the role of T32 in H3.3 vs I32 in H3mm7. Could T32 be phosphorylated? To what extent does the hydrophobic character of I32 disrupt association of H3 tail with the nucleosomal core, decreasing the nucleosome stability? The reciprocal mutant, H3mm7 I32T, should be tested here.

Since we showed that the thermal stability of the nucleosome containing H3.3T32I is unchanged (Fig. 4d), we evaluated the flexibility of the H3 N-terminal tails and the nucleosomal DNA ends. Position 32 is located in the N-terminal tail region of H3. For this purpose, we performed proteinase K and MNase susceptibility assays. We found that there was no notable difference between H3.3- and H3mm7-nucleosomes in either assay (cf. figure below). In addition, the H3.3T32I mutant did not result in a transcription rate change in our updated experiments performed in response to Comment #3. Therefore, we conclude that the H3mm7 I32 residue may not affect the structural and biochemical character of the H3mm7 nucleosome, although it slightly enhanced salt sensitivity through an unknown reason.

MNase assay

Proteinase K resistance assay

All our assays were conducted under the condition that H3 T32 is not phosphorylated. However, the phosphorylation thereof has been reported for plant and animal cells (refs. 1 & 2). It has been reported (ref. 3) that peptides with H3T32ph have a slightly increased amount of interactions with histone chaperones; therefore, H3T32I might induce a change in the status of the interaction with histone chaperones. This was added to the Discussion section as shown below.

As for the reciprocal mutant (H3mm7I32T), this is equivalent to H3.3S57A and we have already studied and reported on it (Fig. 4).

Original:

“H3mm7 and H3.3 had similar patterns of incorporation and histone modification. Therefore, the H3mm7 and H3.3 nucleosomes might be regulated through similar machinery (i.e., histone modification).”

Updated:

“Because H3mm7 and H3.3 had similar patterns of incorporation and histone modification, the observed difference in their functions might be sought in the difference in their amino-acid residues. For instance, the phosphorylation of H3 T32 has been reported for plant and animal cells^{32,33}. Since peptides with H3 T32ph have a slightly increased number of interactions with histone chaperones³⁴, H3T32I might induce a change in the status of such interactions.”

References:

1. Caperta, A. D., M. Rosa, M. Delgado, R. Karimi, D. Demidov, W. Viegas and A. Houben (2008). "Distribution patterns of phosphorylated Thr 3 and Thr 32 of histone H3 in plant mitosis and meiosis." *Cytogenetic and Genome Research* 122(1): 73-79.
2. Tamada, H., N. Van Thuan, P. Reed, D. Nelson, N. Katoku-Kikyo, J. Wudel, T. Wakayama and N. Kikyo (2007). "Chromatin decondensation and nuclear reprogramming by nucleoplasmin (vol 26, pg 1259, 2006)." *Molecular and Cellular Biology* 27(18): 6580-6580.
3. Kumar, A., M. Kashyap, N. S. Bhavesh, M. Yogavel and A. Sharma (2012). "Structural delineation of histone post-translation modifications in histone-nucleosome assembly protein complex." *Journal of Structural Biology* 180(1): 1-9.

3. Is it feasible to examine the T32I and S57A mutants *in vivo*? If yes, this would well connect biological and structural parts. 40 adopts the same conformation and is positioned similarly regardless of the formation of the hydrogen bond with S57. Some more convincing data are needed to make conclusions about the significance of the hydrogen bond between R40 and S57.

To address this comment, we performed RNA-seq to evaluate the effect of H3.3 S57A/T32I in MyoD-forced differentiated NIH3T3 cells as we mention in our response to Comment #1. We added the results for the representative marker genes of skeletal muscle differentiation *Myh3*, *Myog* and *Tnnt2* to Supplementary Figure 23 as shown below. The expression of these genes was enhanced after differentiation for H3mm7 and S57A in comparison with H3.3. These data suggest that A57 in H3mm7 is critical for the change in transcription rates *in vivo*. A discussion on this point was added to the last part of the Results section.

Updated:

“To further assess the enhancement of H3.3S57A on gene expression, we performed RNA-seq for NIH3T3 cells ectopically expressing H3.3S57A/T32I mutants. As expected, the S57A mutant enhanced the expression levels of *Myh3*, *Myog* and *Tnnt2* genes upon differentiation stimuli (representative marker genes of skeletal muscle differentiation, Supp. Figure 23), while the T32I mutant did not.”

<Supp. Fig. 23>

Reviewer #2 (Remarks to the Author):

The authors' research group previously identified a histone H3.3 sub-variant, H3mm7. Here, they tried to investigate the functional roles of H3mm7 during cell differentiation. They claimed that H3mm7 is required for normal skeletal muscle regeneration based on phenotype analysis using H3mm7 knockout mice. Although the crystal structure analysis demonstrate H3mm7-specific A57 residue cannot form a hydrogen bond with the R40 residue of the H4 molecule, the mechanism underlining muscle regeneration deficiency in H3mm7^{-/-} mice remains to be revealed.

I appreciate the reviewer's constructive comments. Our responses to individual comments are given below.

Major points:

1. The authors claimed that H3mm7 is specifically expressed in mouse skeletal muscle tissue (Page 5, lines 93-94). However, when I double checked the reference 14, no convincing data demonstrate such tissue specificity of H3mm7 gene expression.

As the reviewer points out, the mRNA expression of H3mm7 in our previous report (Figure 2b in Maehara & Harada et al., *Epigenetics & Chromatin*, 2015) demonstrated more of a global pattern than a skeletal muscle-specific one. Our intention in this sentence was rather to state our motivation in choosing skeletal muscle as our model of study, namely, that H3mm7 mRNA expression was observed in skeletal muscle tissues and that protein expression was confirmed by mass-spec analysis; furthermore its forced expression enhanced skeletal muscle differentiation. We agree that skeletal muscle specificity of H3mm7 expression is an overstatement and accordingly removed the word "specifically" from the text.

2. In Figure 1A, their data showed that H3mm7 was highly expressed in quiescent satellite cells (SCs) and dramatically decreased after SC activation. In myofibers, expression level of H3mm7 was very low. The data suggest H3mm7 expression is downregulated during myogenic cell differentiation. Thus, how do they think about the observation that deletion of H3mm7 in C2C12 cells delayed cell differentiation (Figure 1D, 1E).

We previously reported that the forced expression of histone H3 variants did not affect the transcriptome in proliferative C2C12 myoblasts, while it did induce changes in gene expression profiles upon differentiation (*Epigenetics and Chromatin* 2015, *Nucleic Acids Res.* 2015). We found, for instance, that the ectopic expression of H3.1 suppressed myogenic differentiation while ectopic expression of H3.3 enhanced myogenic differentiation. In either of these cases, transcriptome analysis detected no significant change in the global gene expression profile, while the chromatin structure was seen to be altered in the undifferentiated stage. These data suggest that a functional change in chromatin structure can take place without an antecedent transcriptional change in undifferentiated cells depending on histone H3 contexts.

To test this hypothesis in H3mm7-knockout cells, we performed ATAC-seq analysis using WT and H3mm7-knockout C2C12 cells analogous to the analysis in Fig. 3 for NIH3T3 cells. The data demonstrated that loss of H3mm7 caused a decrease in chromatin accessibility in proximal gene promoter regions in growth as opposed to WT. We found that the expression levels of the affected genes were lower than those in WT especially after differentiation stimuli. These results suggest that H3mm7 is expressed in satellite cells, but functions to relax chromatin structure to enable induction of gene expression, such as during differentiation. We accordingly added new Supplementary Figure 21 and 22.

When a more advanced low-input epigenomic profiling technology becomes available, we may try epigenome analysis of H3mm7 KO mice. However, due to our limited resources concerning H3mm7 mice, C2C12 cells were the only available choice to address this point.

Updated:

“We additionally performed ATAC-seq analysis using WT and H3mm7-knockout C2C12 cells (Supp. Fig. 21). The data demonstrated that loss of H3mm7 caused a decrease in chromatin accessibility (chromatin accessibility up: 25 genes, down: 137 genes), while the expression levels of the affected genes were decreased only after differentiation stimuli (Supp. Fig. 22). These results suggest that H3mm7 is expressed prior to differentiation and supports the differentiation process.”

<Supp. Fig. 21>

<Supp. Fig. 22>

3. H3mm7 knockout mice exhibit delayed muscle regeneration based on regenerating fiber size and eMHC gene expression. The further experiments should be performed to clarify which cellular event(s) is/are affected by H3mm7 deletion during muscle regeneration: satellite cell activation? Proliferation? Differentiation? Or self-renewal?

We have prepared a new tissue section and analyzed muscle regeneration on day 5 after CTX injury in addition to day 14. We then further performed histological analysis to evaluate the numbers of satellite cells and differentiated cells by staining Pax7 and MyoD after CTX injury. The data showed that the numbers of Pax7⁺/MyoD⁻, Pax7⁺/MyoD⁺ and Pax7⁻/MyoD⁺ cells were not changed in H3mm7^{-/-} mice on day 5 after CTX injury (new Supp. Fig. 4).

<Supp. Fig. 4>

Since the number of satellite cells does not change in H3mm7 KO mice during regeneration after CTX injury, we judged that the population of satellite cells was not affected by loss of H3mm7. Therefore, we performed RNA-seq to evaluate if the expression levels of genes involved in the mentioned regeneration processes (activation, differentiation and proliferation), including skeletal muscle differentiation on days 5 and 14, change in H3mm7 KO. Ideally, with satellite cells such as ASC, QSC and MF are fractionated, various experiments should be performed to address the

comments stated; however, this was not feasible due to our available resources. Therefore, we performed RNA-seq to evaluate the marker gene expression for each cell type (QSC, ASC and MF) that was affected by loss of H3mm7 by comparing expression profiles of wild-type and H3mm7 KO skeletal muscle tissues. RNAs were extracted from tissue sections of WT and H3mm7 KO mice at 5 and 14 days after CTX injury. We found that down-regulated genes in H3mm7^{-/-} mice in contrast to WT included mature muscle fiber-related genes such as *Tnnc2* [ref. 1, 2]. However, the expression of satellite cell specific markers (Fig. 1a) was detected at comparable levels in H3mm7^{-/-} and WT mice, i.e., no remarkable deficiency of gene expression were observed in the SCs (Supplementary Table 1). These results suggest that loss of H3mm7 affects the differentiation process but not satellite cell activation, proliferation or self-renewal.

We appreciate the reviewer's comments which helped further cement our conclusion. We inserted the new sentences as shown below.

Updated:

“We assessed whether the delay in muscle regeneration in H3mm7 knockout mice was caused by a decrease in the number of satellite cells. Immunohistochemistry on day 5 after the induction of muscle injury by CTX showed no notable change in the proportions of Pax7(+)MyoD(-), Pax7(+)MyoD(+) and Pax7(-)MyoD(+) cells around the regenerated muscle (Supp. Fig. 4). We next assessed changes in the gene expression profile upon knockout by RNA-seq (CEL-Seq2) using tissue sections. We fitted the gene expression profile data under injured and uninjured conditions at days 5 and 14 into a statistical model using DESeq2 software (Supp. Table 1). We found that the injury-induced changes to expression levels of satellite cell-related genes were analogous between H3mm7^{+/+} and H3mm7^{-/-} mice. Of these satellite cell-related genes with high fold-changes in *Injured/Uninjured*, *Pax7* demonstrated a log₂ fold-change=3.19 and its expression level was 9-fold higher in the *Injured* condition on days 5 and 14. *Ncam1* similarly showed an 18-fold higher expression level in the *Injured* condition (log₂ fold-change=4.19) and the interaction term for *Injured & Day 14* was -4.16 (one 18th), i.e., the expression level returned to the level in uninjured muscle on day 14. We further observed that down-regulated genes in H3mm7^{-/-} mice included mature muscle fiber-related genes such as *Tnnc2*^{22,23}. These results suggest that the loss of *H3mm7* might affect the differentiation process, while leaving the population of satellite cells unchanged.”

References:

1. Cao, Y., R. M. Kumar, B. H. Penn, C. A. Berkes, C. Kooperberg, L. A. Boyer, R. A. Young and S. J. Tapscott (2006). "Global and gene-specific analyses show distinct roles for Myod and Myog at a common set of promoters." *Embo Journal* **25**(3): 502-511.
2. Di Padova, M., G. Caretti, P. Zhao, E. P. Hoffman and V. Sartorelli (2007). "MyoD acetylation influences temporal patterns of skeletal muscle gene expression." *Journal of Biological Chemistry* **282**(52): 37650-37659.

Ensembl Gene ID	Gene name	Intercept	H3mm7 ^{-/-} /H3mm7 ^{+/+}	Injured/Uninjured	Day14/Day5	Injured & Day14
ENSMUSG000000052155	Acy2a	4.22	-	-	-	-
ENSMUSG00000007655	Cav1	1.92	-	-	-	-
ENSMUSG00000016494	Cd34	1.37	-	-	-	-
ENSMUSG00000031962	Cdh15	1.01	-	2.18	-	-2.25
ENSMUSG00000061353	Cxcl12	2.65	-	-	-1.19	-
ENSMUSG00000045382	Cxcr4	-1.42	-	2.54	-	-
ENSMUSG00000026208	Das	7.78	-	-	-0.71	-
ENSMUSG00000020122	Egr1	-1.55	-	-	-	-
ENSMUSG00000056493	Foxk1	-1.48	-	-	-	-
ENSMUSG00000022484	Hoxc10	3.47	-	-0.81	-	-
ENSMUSG00000027009	Igf1	-	-	1.31	-	-
ENSMUSG00000025348	Igf2	3.46	-	1.58	-	-1.22
ENSMUSG00000025809	Igf1	2.10	-	1.30	-	-
ENSMUSG00000025216	Lbx1	2.77	-	-1.28	-	-
ENSMUSG00000009376	Met	0.81	-	1.14	-	-
ENSMUSG00000009471	Myd1	1.36	-	0.92	-	-2.57
ENSMUSG00000039542	Ncam1	-1.56	-	4.19	-	-4.16
ENSMUSG00000000120	Ngfr	-	-	2.15	-	-
ENSMUSG00000028736	Pax7	-1.58	-	3.19	-	-
ENSMUSG00000025743	Sdc3	-	-	2.70	-	-
ENSMUSG00000017009	Sdc4	1.14	-	-	-	-
ENSMUSG00000000567	Sox9	1.34	-	2.84	1.78	-3.05
ENSMUSG00000027962	Vcam1	-	-	2.63	-	-
ENSMUSG00000032418	Me1	3.52	0.84	-2.70	0.75	1.72
ENSMUSG00000008958	Vps72	3.59	0.77	-	0.93	-
ENSMUSG00000054034	Tcea5	1.92	0.76	1.67	-	-1.29
ENSMUSG0000004980	Hnmpa2b1	3.26	0.69	-	-	-
ENSMUSG00000107002	Og10012G03Rik	3.46	0.68	-0.79	-	-
ENSMUSG00000019970	Sgk1	4.47	0.66	-1.27	-	1.03
ENSMUSG00000008855	Hdac5	3.05	0.65	-	-	-
ENSMUSG00000022635	Zcrb1	3.97	0.62	-1.70	-0.67	1.20
ENSMUSG00000034220	Gpc1	2.07	0.61	0.72	1.52	-1.07
ENSMUSG00000031885	Cbfb	3.36	0.60	-	-	-
ENSMUSG00000063856	Gpx1	1.99	0.59	1.82	-	-
ENSMUSG00000038733	Wdr26	4.45	0.54	0.50	0.58	-0.58
ENSMUSG00000026150	Mif	5.30	0.53	-0.46	-	-
ENSMUSG00000025351	Cd63	3.61	0.53	2.07	0.71	-1.73
ENSMUSG00000001506	Col1a1	2.81	0.53	4.00	1.64	-2.95
ENSMUSG00000030352	Tspan9	5.11	0.47	0.62	-	-
ENSMUSG00000025393	Alp5b	5.10	0.36	-0.50	-	0.56
ENSMUSG00000017300	Tnnc2	8.30	-0.37	-1.10	-0.54	0.92
ENSMUSG00000030647	Ndulc2	5.06	-0.54	-0.71	-0.73	-
ENSMUSG00000008575	Nfib	4.18	-0.54	-	-	-
ENSMUSG00000052305	Hbb-bs	6.10	-0.55	-2.14	-0.87	2.02
ENSMUSG00000024949	Sfl1	3.44	-0.60	0.88	-	-1.04
ENSMUSG00000055322	Tns1	4.50	-0.60	-	-	-
ENSMUSG00000038239	Hrc	3.68	-0.65	-0.80	-	-
ENSMUSG00000027333	Smox	5.35	-0.68	-3.36	-	2.37
ENSMUSG00000053898	Ech1	4.02	-0.72	-1.16	-	1.27
ENSMUSG00000068614	Actc1	8.66	-0.72	3.48	1.86	-3.12
ENSMUSG00000042747	Krtcap2	3.59	-0.73	-	-1.53	-
ENSMUSG00000030433	Sbk2	4.18	-0.93	-1.35	-0.98	2.06

<Supp. Table 1>

4. Next, they tried to understand the mechanism underlining delayed muscle regeneration by RNA-seq. Why do they perform such experiments with C2C12 cells but rather do RNA-Seq using satellite cells from H3mm7^{-/-} mice?

As suggested, we performed RNA-seq with tissue sections prepared for immunohistochemical analysis of muscle regeneration after CTX injury, the data for which is referred to in our response to comment #3 above.

The main reason for our choice of C2C12 cells for RNA-seq was that epigenome analysis requires homogeneous samples of considerable quantity. We agree with the reviewer that ideally all experiments should be done using H3mm7 KO mice, however, the evaluation of epigenomic states using intact satellite cells or tissues was not feasible for us.

5. The authors claimed that H3mm7 might facilitate gene expression by incorporating into nucleosome. After deletion of H3mm7, one should expect the decreased expression level of those H3mm7-regulated genes. How do they think about the RNA-seq data that almost the same number of genes was upregulated in H3mm7^{-/-} C2C12 cells (Figure 2B)? Similarly, why did they observe almost the same number of genes was downregulated in NIH3T3 cells with ectopic expression of H3mm7 (Figure 3B)?

We think that H3mm7 may also function on down-regulated genes upon differentiation; though, in the case of NIH3T3 cells in Figure 3c and new Supp. Fig. 13, H3mm7 preferentially contributed to up-regulating gene expression during differentiation to a greater degree than H3.3. While nearly the same numbers of up- and down-regulated genes were observed in H3mm7^{-/-}, we sought to extract those genes directly associated with H3mm7 regulation of chromatin accessibility from C2C12 cells. Following the reviewer's comment, we performed an ATAC-seq analysis in C2C12 cells as presented in Figure 3c and found that, among the genes where accessibility was diminished (accessibility up: 25 genes, down: 137 genes, as shown in our reply to comment #2 above), some showed lower post-differentiation expression levels compared with those in H3mm7^{+/+} mice (Supp. Fig. 22). These results suggest that H3mm7 may have a more robust effect in the up-regulation of gene expression. We are grateful for the reviewer's comment which helped further solidify this result.

We accordingly added the breakdown of up- and down-regulated genes among the 3,505 genes (up: 1,670, down: 1835) in Figure 2a and new supplementary figures as explained above (comments #2 and 3).

Reviewer #3 (Remarks to the Author):

In their manuscript "Histone H3.3 sub-variant H3mm7 is required for normal skeletal muscle regeneration" Harada et al. investigate the specific role of the recently identified histone variant H3mm7 on muscle regeneration in mouse. They find that H3mm7 is specifically expressed in satellite cells and is necessary for proper muscle regeneration after injury in vivo and also for muscle

differentiation in vitro. Their data suggest that presence/absence of H3mm7 may specifically affect the expression of genes upregulated during muscle differentiation. This could be due to the fact that nucleosomes containing H3mm7 are less stable than H3.3 containing nucleosomes and would therefore favour transcription of a subset of genes.

I find the paper original and interesting in the context of histone variants and their specific roles in chromatin activity. The authors use several approaches offering different angles to study functional differences between H3mm7 and H3.3, which differ from each other only by two amino acids. It is rather clear that H3mm7 depletion impair muscle regeneration in vivo and differentiation of C2C12 cells in vitro. I think also that the data support well a model where nucleosomes containing H3mm7 are less stable than H3.3 containing nucleosomes. I found however more difficult to appreciate the specific role of H3mm7 on gene expression during differentiation. I feel that more data and robust statistical analysis should be included in the manuscript to firmly conclude that H3mm7 expression modulates the transcriptional activity of genes that are differentially expressed across differentiation and to support a role distinct from H3.3.

We appreciate the reviewer's careful reading of our manuscript, which caught various mistakes which would otherwise have gone unnoticed. We are grateful for his comments and suggestions which were instrumental in improving the quality of our work. We have performed additional experiments and data analysis to address the concerns expressed.

Major points

1. Supplementary Fig4: there is no statistical analysis. I think this should be done considering the variation observed for some time points, in particular in the H3mm7^{-/-} cells. There is also a difference between the number of "measurements" between the different conditions: for example, 14 points at 48h for H3mm7^{-/-} vs 7 points for H3mm7^{+/+} at the same time point. The standard deviation bars are also a bit surprising: how can it be so small in the case of H3mm7^{-/-} cells at 48h with such a distribution of values?

We addressed these three points as described below.

<Supp. Fig. 6>

(1) Statistical analysis

We added the results of our statistical tests to the figure.

(2) Varying numbers of measurements

We agree that the varying numbers of measurements were inappropriate. We re-calculated the fusion-index and performed statistical tests (at 48 and 72 h) using 14 measurements.

(3) Small error bars

The error bar in Supp. Fig. 4 is the standard error (standard error of the mean, SEM) as we stated in the legend, and not the standard deviation (SD) of the measurements. The standard error is the standard deviation of the sample mean, and is inversely proportional to the square root of the sample size (N), and hence the widths of the error bars are always smaller than SD.

We incorporated these points in the updated Supplementary Figure 6 and the legend.

2. Supplementary Fig5: the western blot is not convincing. A picture showing a lower exposure for H3 should be shown (where the level of H3 for each lane could be clearly seen) and a clean immunoblot should be shown for H2B (or no blot at all and remove the statement on H2B levels in the main text since it does not add anything to the main conclusion). Since the levels of Hsp90

detected in the 4 lanes are different, it is important to show clear immunoblots for H3 and quantification of the signals (related to Hsp90) may be useful here.

To address this point, we repeated the western blotting and evaluated the band intensities to determine the ratio of the total H3 level to the level of Hsp90. The data showed that the total H3 level was nearly constant among the cells. The original western blot data were replaced with the updated data as shown.

<Supp. Fig. 7>

3. Page 7, lines 146-147: More genes would be necessary to support the conclusion of the authors. Indeed, one gene only (*Ckm*) is fully supporting a function unique to H3mm7 in the regulation of genes associated to skeletal muscle differentiation. The other gene (*Des*) however does not support this conclusion.

Following this comment, we added three more genes to the quantitative RT-PCR analysis. As the reviewer pointed out, not all myogenic genes were significantly affected by the H3mm7 KO condition as we showed in the case of *Des* ($p=0.064$). All of the three genes which we newly tested were confirmed to be rescued by H3mm7. Based on these results, we updated the text as shown below.

Updated:

“However, the level of *Des*, expressed prior to differentiation, was rescued by H3.3 ($p = 0.032$), which was not the case for H3mm7 ($p = 0.064$). The levels of the control (housekeeping) gene *Eef1a1* were not significantly changed in *H3mm7*^{+/+}, *H3mm7*^{-/-} or the rescued clones of H3mm7, although an elevated level of *Eef1a1* was observed upon H3.3-GFP expression. Taken together, while not all genes that were down-regulated by the loss of the H3mm7 gene were rescued by the ectopic expression of H3mm7-GFP, a group of myogenesis-related genes were observed to be significantly rescued (Fig. 1e and Supp. Fig. 8), suggesting that H3mm7 might have a function to regulate these

genes. These results suggest that expression of H3mm7 might affect upregulated genes in differentiation.”

<Supp. Fig. 8>

4. Page 10, lines 200-201: The data supporting the conclusion are not strong enough. Indeed, only two replicates were done for each cell type (and each time point) and the two replicates for the H3mm7+/+ show variations that make the interpretation of the results difficult. This is particularly the case for the house keeping genes (showing no variations across the differentiation process) that were detected as DEGs by DESeq2. How to be sure that H3mm7 depletion has really no effect on the expression of these genes and only modifies the expression of genes associated with differentiation? To support the idea that H3mm7 only regulates the expression of genes affected by the differentiation process, a third replicate should be included in the analyses (at least for H3mm7+/+ cells) to verify that housekeeping genes expression is not altered in absence of H3mm7 and that the results regarding the other genes are not due to variations between replicates.

Since we have duplicates for each of the eight conditions (four time points with two genotypes), in the comparative analysis of WT/KO conditions (i.e. testing for the rate change phenomenon discussed in the manuscript), this amounts to having eight replicates for each test (i.e., 11 degrees of freedom).

Moreover, to evaluate statistically whether the current number of replicates suffices for the detection of DEGs, we performed a post-hoc power analysis on our DESeq2 results using the R package *locfdr* of Efron et al. The result showed that the statistical power in WT vs. H3mm7-/- was 67% at the threshold of “local FDR” < 0.2 (corresponding to FDR < 0.1 in this work), which can be interpreted to mean that 67% of the potential DEGs were detected (for reference, the result of the power analysis is shown below). Thus, since the greater part of the non-null (significant) genes were already detected in the current setup, we judged that additional replicates were unnecessary.

Reference:

Efron, Bradley. "Size, Power and False Discovery Rates." *The Annals of Statistics* 35, no. 4 (2007): 1351-1377.

5. Supplementary Fig11: The calculation of a difference between mean values is a weak method to support the fact that H3mm7 overexpression has more effect on gene expression than H3.3. The data showed in the right panel should at least contain standard deviations. What test was used to determine the p-values in that particular case?

<Supp. Fig. 14a>

We agree that a difference in means would not be adequate; however, what we evaluated here is the mean of differences (log-ratios). The p-values were calculated by two-sided t-test (null hypothesis: the difference is 0) after Bonferroni correction for multiple comparisons. As stated in the legend, we also presented 95% confidence intervals as error bars with regard to the t-distribution (which we admit are

very narrow and not clearly visible in places) for the visualization of where it is significant (i.e., where the difference does not equal zero), namely, where the intervals do not intersect the horizontal line $y=0$. We added explanation of the statistical test to the legend of new Supplementary Figure 14.

6. Page 10, lines 216-218 (and Supplemental Fig11): To show that "the expression of H3mm7 led to more rate change [...] compared with that caused by H3.3" the authors should compare the ratios H3mm7/control and H3.3/control (in terms of gene expression, for each gene) and verify whether or not they are globally, statistically different. The data are not strong enough as they are presented to support the conclusion.

As explained above, we performed a statistical test on the data in old Supp. Fig. 11 on the mean of differences in individual gene expression, and thus the existence of a significant difference between the groups was statistically confirmed. Following the reviewer's comment, we added the comparisons against control NIH3T3 cells.

<Supp. Fig. 14b>

Updated:

“We also performed an analysis analogous to that in Supplementary Figure 14a against control (Supp. Fig. 14b). The results showed that both H3mm7 and H3.3 cases demonstrated significant mean differences against WT in all the clusters, which suggests that the ectopic expression of H3.3 and H3mm7 have analogous effects on the gene expression profile. However, the difference was not significant at 24 h in the clusters UA and DA in the case of H3.3, and H3mm7 induced greater rate changes in comparison with H3.3; these facts might lead to the H3mm7-specific gene expression profile.”

While the study of the effect of H3.3 itself on the gene expression profile is of great interest to us, we judged that a detailed analysis of it would not serve the purpose of the paper, which is the functional analysis of H3mm7. However, we would like to come back to this theme in future work.

7. Page 11, lines 222-223 and Supplementary Fig12: the qPCR results are actually not fully supporting the RNA-seq data. Indeed, in the case of *Ckm*, H3.3 and H3mm7 seem to have the same effect on gene expression. In the case of *Myh3*, it is even more surprising since at 48h, H3mm7 expression has no significant effect on gene expression while H3.3 has. To which cluster does *Myh3* belong (my guess is UA)? Regarding the expression of this gene, overexpression of H3mm7 and H3.3 appear to have opposite effects, which is not the case in any of the clusters shown in Fig3B. Again, more genes here would be necessary to get a better picture of the effect of H3mm7 on gene expression compared to H3.3 in NIH3T3 cells.

Following the reviewer's suggestion, we additionally performed qPCR for four more genes involved in skeletal muscle differentiation and compared the results with the RNA-seq data (updated Supp. Fig. 15). These genes, along with *Myh3* and *Ckm*, all belong to the cluster UA. We agree with the reviewer that the forced expression of H3.3 and that of H3mm7 had analogous effects on *Ckm*. The results of qPCR agreed with the RNA-seq data, namely, there were genes enhanced only by H3mm7 (such as *Des*, *Tnni2* and *Myh3*; H3.3 even induced down-regulation of *Myh3*) and those by both H3mm7 and H3.3 (such as *Acta1* and *Tnni2*) found in cluster UA. However, the cluster mean values in Fig. 3b show that a greater part of cluster UA consists of genes where the enhancement effect was greater in the case of H3mm7 compared with WT, while there seem to be minority genes which follow the UA pattern and yet are negatively affected in expression by the ectopic expression of H3.3. The study of H3.3-specific gene expression regulatory functions is of great interest to us, and we consider it a future agenda. Following the discussion above, we updated the text as follows

<Supp. Fig. 15>

Original:

“We confirmed mRNA-seq data by qPCR for the differentiation marker genes, *Ckm* and *Myh*. Although Figure 1d and Supplementary Figure 4 show that H3.3 only partially compensated for the function of H3mm7, these results identified a regulatory function of the rates of expression levels specific to H3mm7.”

Updated:

“We confirmed RNA-seq data by qPCR for the representative differentiation marker genes belonging to cluster UA (Supp. Fig. 15). While the plot in Figure 3b showed notable rate changes in differentiation induced by H3mm7, the genes in Supplementary Figure 15 included those for which the enhancement effect was greater in H3mm7 (*Des* and *Tnnt2*), those in which H3.3 and H3mm7 have comparable enhancement effects (*Acta1*, *Ckm* and *Tnni2*), and a group of genes including *Myh3* for which the ectopic expression of H3.3 resulted in lower expression levels compared with WT. The results in Figure 1d and Supplementary Figure 6 show that H3.3 only partially compensated the function of H3mm7; however, the results of RNA-seq analysis in NIH3T3 cells also indicate that H3mm7 has a greater impact on gene expression than H3.3.”

8. Page 11, lines 238-241: Another obvious conclusion from this part is that H3.3 and H3mm7 are deposited overall at the same places within the genome. This goes in the direction of overlapping functions between the 2 variants.

As the reviewer points out and as we discussed above, we found that, from the analyses of the current work and our Epigenetics & Chromatin paper, H3.3 and H3mm7 have much in common in their localizations and effects on the gene expression profile. One of the theses of the current work is that the unique functions of the sub-variants stem from the subtle change in their functions after their incorporation. We updated the conclusion of the section as follows.

Original:

“Therefore, this suggests that the regulation of expression rate by H3mm7 incorporation did not originate from the specific change of histone modification or incorporation sites on the genome.”

Updated:

“Therefore, this indicates that the regulation of expression rate by H3mm7 incorporation did not originate from the specific change of histone modification or incorporation sites on the genome. Despite the fact that H3.3 and H3mm7 are nearly identical in their sequences, localization and histone

modification preferences, the rate change effect of H3mm7 made a notable difference in the gene expression profile after differentiation.”

9. Page 12, lines 252-253 and Fig3C: There are no statistics or numbers to support that "the chromatin accessibility and expression of the genes belonging to cluster UA were enhanced with H3mm7 compared with H3.3". Moreover, the figure only shows the difference in gene expression between H3mm7 expressing cells and control cell without overexpression of H3.3. Fig3B addresses this point so if the authors conclusion is also based on panel B, they should refer to it as well.

To address the reviewer’s suggestion, we added new analysis and interpretation of Figure 3b (and old Supp. Fig. 11, now in Supp. Fig. 14) to support the conclusion that "the chromatin accessibility and expression of the genes belonging to cluster UA were enhanced with H3mm7 compared with H3.3." Of the genes which showed significant changes in the ectopic expression of the histone variant ($|\log_2FC| > 1$ in H3mm7 vs. WT), we noted the numbers of those in each quadrant for each cluster in Supplemental Figure 20. We also added references to Fig. 3b and H3.3 vs H3mm7 in new Supp. Fig. 14.

Original:

“The chromatin accessibility and expression of the genes belonging to cluster UA were enhanced with H3mm7 compared with H3.3.”

Updated:

“The enhancement effect of H3mm7 on gene expression was greater than that of H3.3 (Figure 3b and Supp. Fig. 14); therefore, we compared the numbers of genes belonging to cluster UA in quadrants I and IV (i.e., greater enhancement effect both in chromatin accessibility and gene expression in H3mm7+ cells vs. greater enhancement effect in H3.3+ cells). We found the result to be I: 39 vs. IV: 9, and hence we confirmed that there were more genes with enhancement levels greater by more than 2-fold-changes in chromatin accessibility and gene expression induced by the ectopic expression of H3mm7 (Supp. Fig. 20). Therefore, we concluded that genes belonging to cluster UA were enhanced more in both accessibility and expression by H3mm7 than by H3.3.”

<Supp. Fig. 20>

10. Page 12 lines 260-262: There is so far no strong evidences that H3mm7 has (i) a specific role in the expression of genes related to myogenic differentiation and (ii) compared to H3.3.

As we have been discussing above, it was not our intention in this sentence to assert that the rate change effect is H3mm7-specific, but rather that the enhancement of gene expression rate by H3mm7 in cluster UA was more prominent than that by H3.3. Following the reviewer’s suggestion, we emphasized this point in the text as follows.

Original (page 12 lines 260-262):

“Together, these results suggest that the change of chromatin accessibility at H3mm7-incorporated gene expression rate enhances the upregulation of the expression of myogenic-differentiation-related genes.”

Updated:

“Together, these results suggest that the change of chromatin accessibility at H3mm7-incorporated genes enhances the upregulation of the expression of myogenic-differentiation-related genes notably in cluster UA to a greater degree compared with H3.3+ cells.”

11. Page 13, line 284 – page 24, line 287 and Fig4E: in which cell type was the FRAP experiment done? More cells would have made the conclusion stronger. The authors are mentioning a faster turnover of H3mm7 and H3.3S57A. However, looking and the panels showing the photobleaching (0 min) it appears that the bleaching is incomplete in the case of H3.3S57A and to a lower extent with H3mm7. This could be due to the presence of an insoluble pool that is not incorporated rather than just rapid turnover (due to instability). This could be tested by doing a simple Triton TX-100 extraction to separate the soluble and insoluble fractions. A western blot showing the level of

expression of the different exogenous proteins should also be done to verify that the presence of a soluble pool is not due to a high overexpression.

We have analyzed more cells and replaced the graph with the new data which are analogous to those obtained by the small number of cells originally presented. As the reviewer pointed out, the intensity of the bleached area just after bleaching is higher for H3.3S57A and H3mm7, indicating the presence of relatively larger diffusible (non-incorporated) fractions in these cells. This is actually consistent with instability of H3.3S57A and H3mm7 in nucleosomes, given that the fluorescence intensities (i.e., the expression levels) were similar among all H3 proteins and that the curve shapes for H3.3S57A and H3mm7 are steeper than those for H3.3 and H3.3T32I.

We accordingly updated the text as follows.

Original:

“The fluorescence recovery rate at 240 min after photobleaching was less than 20% for H3.3 and H3.3T32I nucleosomes but was 40%–50% for H3mm7 and H3.3S57A.”

Updated:

“After photobleaching, the fluorescence intensity of GFP-H3.3 and GFP-H3.3T32I dropped to ~10% of the original intensity and recovered to ~20% in 240 min. The intensity just after bleaching was slightly higher for GFP-H3mm7 and GFP-H3.3S57A, and recovered up to ~50%-60%, indicating that these proteins exchange more rapidly than GFP-H3.3 and GFP-H3.3T32I.”

Minor points

In the RESULTS section:

1. It is difficult to understand how the CEL-Seq2 experiment was conducted (page 5, lines 98-104). First the author say that "global gene expression [...] was performed at different stages of myogenesis using quiescent or activated satellite cells and myofibers isolated from the extensor digitorum longus (EDL) muscle". Then they refer to a muscle injury induced by injection of CTX into the gastrocnemius muscle in a certain type of mice to explain the origin of the cells they mentioned earlier. The procedure used should be put before the results referring to Fig1A.

We revised the paragraph in question as follows to clarify the source of the data.

Original:

“Hence, global gene expression analysis using CEL-Seq2¹⁷ was performed at different stages of myogenesis using quiescent or activated satellite cells and myofibers isolated from the extensor digitorum longus (EDL) muscle (Fig. 1a and Supplementary Fig. 1 for the correlation coefficients matrix of each sample). Muscle injury was induced by cardiotoxin (CTX) injection into the gastrocnemius muscle in Pax7-CreERT2;Rosa26-YFP mice, and then Pax7 (YFP)-expressing quiescent satellite cells or activated satellite cells were isolated from uninjured or injured muscle, respectively (See Methods for details). We found that *H3mm7* was synchronously expressed with marker genes in activated (*Pax3/7+*, *MyoD1+*) and quiescent SCs (*Pax3/7+*, *Calcr+*)¹⁸.”

Updated:

“Hence, global gene expression analysis using CEL-Seq2¹⁷ was performed at different stages of myogenesis. We prepared SCs by muscle injury induced by cardiotoxin (CTX) injection into the gastrocnemius muscle in Pax7-CreERT2;Rosa26-YFP mice, and then Pax7 (YFP)-expressing cells

were isolated as SCs (quiescent SCs from uninjured muscle and activated SCs from injured muscle). Myofibers were isolated from the extensor digitorum longus (EDL) muscle (see Methods for details). We found that *H3mm7* was preferentially expressed in satellite cells (activated: *Pax7+*, *Myod1+* cells and quiescent: *Pax7+*, *Calcr+* cells)¹⁸ compared with myofibers (Fig. 1a and Supplementary Fig. 1 for the correlation coefficient matrix of each sample).”

2. Page 5, line 105: *Pax3* is not expressed in activated satellite cells according to Fig1A so *H3mm7* should not be mentioned as synchronously expressed with *Pax3* in these cells (but with *Pax7*, yes).

Based on the comment, we revised the text as above (removed *Pax3* and the word “synchronous”).

3. Which satellite cells (activated or inactivated) were used to assess the cell-specificity of *H3mm7* expression by RT qPCR in Supplementary Fig2? It is not indicated in the text, figure legend or methods.

We apologize for the lack of explanation. We used a *Pax7* positive fraction expanded *in vitro* as a source of satellite cells, of which the greater part should be ASCs (see Fig. 2, and as previously demonstrated in Fujimaki et al, Stem Cells., 2017). To perform allele-specific qPCR, which requires a substantial amount of input RNA, we used this fraction in Fig. 2. We added an explanation of the definition of “SC” to the legend.

4. In Supplementary Fig2, why is *H3mm7* expression normalized to *H3f3a* and not *Gapdh* as for *H3f3a* and *H3f3b*? *H3mm7* expression normalized to *Gapdh* should be shown as well.

We were not able to detect the expression level of *H3mm7* using standard RT-qPCR. To overcome this problem, we applied allele-specific PCR (Zhang et al., J-Neuro Oncol., 2016) to compare the levels of highly similar sequences. Hence, we did not compare the level of *H3mm7* with that of *Gapdh*, but rather with that of *H3f3a*.

5. Supplementary Fig2: The legend should be re-written.

Based on this comment, we updated the legend and method “Variant-specific Quantitative RT-PCR” according to the comments #3 and 4 above.

6. Page 5, lines 108 – 110: Why the fact that H3f3a also shows a decreased expression in myofibers participates to the conclusion "H3mm7 functions in earlier stages of skeletal muscle regeneration".

We agree that the said decreased expression of H3f3a is relevant for our conclusion here, and we accordingly removed the reference to H3f3a in the text as shown below.

Original:

“As H3f3a and H3mm7 both showed decreased expression in myofibers (MFs) ($p=0.044$ and $p=0.006$), we considered H3mm7 functions in earlier stages of skeletal muscle regeneration.”

Updated:

“H3mm7 showed decreased expression in myofibers (MFs) ($p=0.006$); therefore, we considered H3mm7 to function in earlier stages of skeletal muscle regeneration.”

H3f3a's function in earlier stages of skeletal muscle regeneration is naturally of interest, because it is also expressed there. However, we are of the opinion that it would be out of topic in this study and should be addressed in future work.

7. The "H3mm7 knockout mouse line" is not documented at all. The authors should at least include a figure showing that these mice lack the H3mm7 gene.

Following this comment, we added the genotyping data to the new Supplementary Figure 3.

8. Fig1D: the Hoechst staining is almost impossible to see in the upper panels. It is thus very difficult to visualize the undifferentiated cells, especially among the H3mm7^{-/-} cells (top-right). Showing cells with higher magnification would strengthen the conclusion.

We apologize for the lack of clarity, which was due to the low resolution of the data. We replaced the image with one having higher resolution along with another representative image as an intuitive sample (the updated Figure 1d).

9. Page 6, lines 113-115: "H3mm7 knockout mice are viable" is a weak argument for stipulating that "H3mm7 is not necessary during development". If the authors mean that these mice are indistinguishable from control mice, they should mention it instead. Indeed, "being viable" does not

mean to be "normal" and these mice could suffer from various malformations, which does not seem to be the case.

We agree that "being viable" does not mean "normal." We updated the text as follows.

Updated:

"H3mm7^{-/-} mice were viable and indistinguishable from control mice"

10. Page 6, line 117: a reference to Fig.1B should be put after "[...] CTX injection".

We are grateful to the reviewer for pointing this out, and we duly moved the reference.

11. Page 6, line 118: a reference could be added at the end of the sentence (regarding expression of dMHC during muscle regeneration).

We accordingly added references regarding the expression of dMHC during muscle regeneration. We are grateful for the advice.

12. Fig1B: the red signal is weak and could be brighter so it would be easier to see.

We duly updated Fig. 1b.

13. Page 6, line 128: a reference to Fig1D (lower panels) could be added at the end of the sentence. I would also specify "ectopic expression of ..." instead of just using "H3mm7-GFP".

We agree and updated the sentence according to the reviewer's comment.

14. Page 6, line 131: the authors do not discuss at all why ectopic expression of H3.3 could worsen the phenotype observed in H3mm7^{-/-} cells. This is an interesting observation that should be discussed.

We agree that such an observation would be interesting, but such effects might not be global on myogenic genes because they were not manifested as a specific cluster in our RNA-seq analysis (Fig. 2) and so we did not pursue the avenue further. However, we added the following sentence to the text.

Updated:

“H3.3 did not rescue the expression levels of *Ckm* at 48 h, but rather caused a decrease in expression levels of some myogenic genes ($p = 0.003$; lower than *H3mm7^{-/-}*).”

15. Page 6, line 132: What is "the H3-C antibody"? Even though the reference (44) is indicated in the methods, it should be mentioned here as well.

We agree and updated the text as follows.

Original:

“We also evaluated the total amount of histone H3 using the H3-C antibody”

Updated:

“We also evaluated the total amount of histone H3 in C2C12 cells using an antibody that recognizes the C-terminus of histone H3 (anti-H3-C; see Methods for details)”

16. Page 6, lines 131-133: The cell types that are compared should be mentioned.

We apologize again for the lack of explanation. These were C2C12 cells as in Fig. 1. We added an explanation in the revised text as shown above.

17. Page 7, lines 137-138: *Eef1a1* is not a muscle marker, as mentioned later (used as a control, housekeeping gene). Thus only 2 representative skeletal muscles markers are used.

We thank the reviewer for the advice and updated the text accordingly as follows.

Updated:

“... the representative skeletal muscle markers, *Ckm* and *Des*, and the housekeeping gene, *Eef1a1* as a control (Fig. 1e).”

18. Fig1E: What are the orange and blue bars? It is not indicated in the figure or in the figure legend. Also, the reference bar corresponding to expression at 48h in H3mm7^{+/+} cells (=1) should also show a standard deviation. To do so, each value of the reference sample should be divided by the mean value for this sample. With the resulting values, one can get a standard deviation centred on 1.

We apologize for the loss of visual cues. Blue bars signify expression levels at the growth state and orange those at 48 h after differentiation stimulus. We updated Figure 1e.

19. Page 7, lines 137-138: The authors write that "the mRNA level of *Ckm* was decreased in H3mm7^{-/-} cells at 48 h after the differentiation stimulus". This sentence is confusing since it may indicate a decrease of *Ckm* expression upon induction of differentiation, and that is not the case. I think the authors should re-write this sentence.

Based on the comment, we updated the text as follows.

Updated:

“The mRNA level of *Ckm*, which is highly expressed in differentiation, was lower in H3mm7^{-/-} cells than in H3mm7^{+/+} cells at 48 h after the differentiation stimulus ($p = 0.001$; lower than H3mm7^{+/+}) and was rescued by H3mm7 expression ($p = 0.001$; higher than H3mm7^{-/-}).”

20. Page 7, lines 142-147: The authors claim that "H3.3 and H3mm7 both rescued *Des* expression", relying on p values of 0.064 and 0.032 respectively, even though Fig1E shows no statistical differences in case of "H3mm7^{-/-} +H3mm7". However, they write just after that "levels of the control [...] gene *Eef1a1* were not significantly changed in H3mm7^{+/+}, H3mm7^{-/-} or the rescued clone" while Fig1E shows a statistically significant increase of expression in the "H3mm7^{-/-} + H3.3" experiment. This part should be re-written taking into account the statistical significance, or not, of the differences.

We thank the reviewer for pointing out the inconsistency in our explanation. We updated the sentence alongside the results of the statistical test. Please see below.

Original:

“*Des* was increased after differentiation; however, H3.3 and H3mm7 both rescued *Des* expression levels at 48 h ($p = 0.064, 0.032$; higher than *H3mm7*^{-/-}). Levels of the control (housekeeping) gene *Eef1a1* were not significantly changed in *H3mm7*^{+/+}, *H3mm7*^{-/-} or the rescued clones.”

Updated:

“However, the level of *Des*, expressed prior to differentiation, was rescued by H3.3 ($p = 0.032$), which was not the case for H3mm7 ($p = 0.064$). The levels of the control (housekeeping) gene *Eef1a1* were not significantly changed in *H3mm7*^{+/+}, *H3mm7*^{-/-} or the rescued clones of H3mm7, although an elevated level of *Eef1a1* was observed upon H3.3-GFP expression. Taken together, while not all genes that were down-regulated by the loss of the H3mm7 gene were rescued by the ectopic expression of H3mm7-GFP, a group of myogenesis-related genes were observed to be significantly rescued (Fig. 1e and Supp. Fig. 8), suggesting that H3mm7 might have a function to regulate these genes. These results suggest that expression of H3mm7 might affect upregulated genes in differentiation.”

21. Supplementary Fig6: I am not familiar with DESeq2 and I cannot evaluate the method. However, I have the feeling that a Y-axis legend is missing to better understand the cartoon. Is it gene expression level?

That is correct, and we accordingly added a label for the y-axis.

22. Page 8, lines 159-162: "Loss of gene expression [...]": Why is this sentence here? Is it to show that the method used by the authors (DESeq2) is working in other contexts? If so, the author should at least say it.

We thank the reviewer for the suggestion. We inserted the text before the explanation of our statistical model as follow.

Updated:

“Loss of gene expression kinetics is commonly detected in the gene knockout or knock-down models as representatively shown in Supplementary Figures 10 and 11 for si-mH2A.1 in C2C12 cells and for BMAL KO in hip cartilage samples, respectively. In the latter study, the authors utilize DESeq2’s likelihood ratio test in a setup analogous to ours (*Time* and *Phenotype* factors) to demonstrate that a group of genes lose the periodical pattern of expression. Following their method, we employed a statistical model using DESeq2 ...”

23. Supplementary Fig7 : What is MB? MT? cond? si-mH2A? Is mH2A.1.2i the same thing as si-mH2A? Why are the points corresponding to the same time point not all aligned (y-axis). Is it normal?

We apologize for the unexplained abbreviations. We added abbreviation definitions to the legend. We also slid the plots in the direction of the x-axis so that error bars at same time points do not overlap.

24. Supplementary Fig8 : What is cond? Same as for Supplemental Fig7.

We updated the legends as explained above.

25. Page 8, line 167: Why "even"?

Our intention here was to suggest the possibility that the C2C12 H3mm7^{-/-} cells, despite having the equivalent genomic structure, can show different expression profiles at the single-cell level. We admit that the message was not well delivered with the word “even” as the reviewer points out, and accordingly revised the text as follows.

Original:

“even in the isogenic H3mm7^{-/-} C2C12 cell line” (line 170)

Updated:

“despite the isogenicity of the H3mm7^{-/-} C2C12 cells”

26. Page 9, lines 180-181: The way the conclusion is written tends to suggest that H3mm7 is not specifically involved in the expression of genes related to myogenic differentiation. Is it what the authors mean?

That is correct; we updated the text following this suggestion and Comment #20.

Original:

“These results suggest that the rate change of transcription caused by H3mm7^{-/-} affected both myogenic and non-myogenic genes”

Updated:

“These results suggest that, while H3mm7 is not involved specifically in the expression of genes related to myogenic differentiation, it may be required to upregulate gene expression upon differentiation.”

27. Fig2B: the legend on the right side of the graphs (under cluster) shows SA and SB clusters, while this is not shown in any graph. This should be removed.

We removed the labels SA and SB.

28. Page 9, line 188 : "Flat staying patterns"? What is this?

We agree that “flat staying patterns” is rather ambiguous. We changed it to “flat patterns (no significant change in time; cluster SA and SB)”.

29. Supplementary Fig9 : What is the y-axis and where are the values? Where is the legend for the symbols used in the graphs? What is G? Why using D24-72 in this figure while using 24-72h in other figures?

We apologize for the insufficient explanation. We updated the figure (new Supp. Fig. 12) to match the labels of the main figures.

30. Page 9, lines 191-192: The conclusion is confusing. "The rate change of gene expression" between what and what? "Up- or down-regulation of gene expression" occurring when? The authors juggle the concepts of (i) gene regulation across differentiation and (ii) differentially expressed genes between H3mm7^{+/+} and H3mm7^{-/-} cells. This makes the reading difficult.

We are grateful for the reviewer’s advice and accordingly updated the text as shown below.

Original:

“These data indicated that the rate change of gene expression occurred with up- or down-regulation of gene expression.”

Update:

“These data indicated that loss of the *H3mm7* gene caused the rate change in expression of up- and down-regulated genes upon differentiation.”

31. Page 9, line 193 : "We next evaluated the effect of rate change...". The effect on what?

We apologize for the confusion which our inappropriate expression (“the effect”) might have caused. In the paragraph directly before, we gave our interpretation of the k-means clustering results as “these data indicated that the rate change of gene expression occurred with up- or down-regulation of gene expression.” To show that this is unbiased, we performed linear discriminant analysis (LDA). We revised the text as shown below.

Original:

“We next evaluated the effect of rate change of gene expression cause by loss of H3mm7. Fisher’s liner discriminant analysis (LDA) was performed to extract the component to explain the clustering presented in Figure 2b (Fig. 2c). [...] This suggested that rate change was the major component to explain the result of k-mean clustering. Therefore, H3mm7 is suggested to be a factor that regulates the expression rates of genes affected by the differentiation process.”

Updated:

“We further confirmed that the rate change had occurred with up/down-regulation of gene expression after the loss of *H3mm7* by Fisher’s linear discriminant analysis (LDA). LDA was used to estimate the superpositions of the expression patterns in the clusters of Figure 2b (Fig. 2c). The singular values (Fig. 2d) representing the performance of the class separation showed the two major components, LD1 (1st component of LDA; corresponding to A and B) and LD2 (U, D) in contrast to the other minor ones (LD3, 4 and 5). The singular values of LD1 and LD2 were 123.98 and 50.07 (84.3% and 13.8% of total between-class variance), respectively. This suggested that the rate change (bias) between *H3mm7*^{-/-} and *H3mm7*^{+/+}, and subsequently the up/down-regulations in differentiation, were indeed the two major components that explain the result of k-means clustering.”

32. Fig2D: what are LD3, LD4 and LD5? This is not said anywhere.

The LD_i's are the i-th extracted components of Fisher’s linear discriminant analysis (total of k-1 components exist for the k-class problem; k=6 in our case). We updated the main text as follows.

Original (page 9, line 197):

“The singular values (Fig. 2d) representing the performance of the class separation showed the two most reasonable components, LD1 (corresponding to A, B) and LD2 (U, D).”

Updated:

“The singular values (Fig. 2d) representing the performance of the class separation showed the two major components, LD1 (1st component of LDA; corresponding to A and B) and LD2 (U, D) in contrast to the other minor ones (LD3, 4 and 5).”

33. Supplementary Fig10 : What chromosome is shown on the figure? The Y-axis has no label or legend.

We appreciate the reviewer’s careful reading. We added the information of the scale (depth) for the y-axis and loci as shown in the updated figure (Supp. Fig. 13) and modified the label for a better description of the data. The snapshot shows the results of the amplicon sequences of the gene loci around *H3mm7*.

<Supp. Fig. 13>

34. Page 10, line 207: " [...] GFP-fused H3.3/H3mm7 NIH3T3 cells...". This should be better explained.

Following this comment, we added a discussion on the lack of the *H3mm7* gene locus in NIH3T3 cells as follows: “We confirmed that NIH3T3 fibroblast cells lack the *H3mm7* gene locus by amplicon sequences (Supp. Fig. 13).”

Original:

We performed time-course RNA-seq with GFP-fused H3.3/H3mm7 NIH3T3 cells that naturally lack the *H3mm7* gene locus (Supplementary Fig. 10).

Updated:

“We confirmed that NIH3T3 fibroblast cells naturally lack the *H3mm7* gene locus by analysis of amplicon sequences (Supp. Fig. 13). Hence, we performed time-course RNA-seq in NIH3T3 cells with GFP-fused H3mm7 and with H3.3 as control.”

35. Page 11, lines 223-225: the way the conclusion is written emphasizes a role of H3.3 in gene regulation instead of minimizing it. The sentence should be re-written. And again, I am not convinced that the effect of H3mm7 expression in NIH3T3 cells differs globally from H3.3 overexpression, for the reasons mentioned above.

As the reviewer points out, the rate change might be common in H3mm7 and H3.3. However, the results in Supplementary Figure 14 (old Supp. Fig. 11) indicate that the rate change effect per H3mm7 is more significant than that of H3.3. This disparity is what was here referred to by the word “specific,” and we concluded that this consequently resulted in the enhancement of skeletal muscle gene expression from our GSEA results. It was not our intention to state that H3.3 has no rate change effect. Following the reviewer’s point, we updated the text as shown below.

Original:

“Although Figure 1d and Supplementary Figure 4 show that H3.3 only partially compensated for the function of H3mm7, these results identified a regulatory function of the rates of expression levels specific to H3mm7.”

Updated:

“The results in Figure 1d and Supplementary Figure 6 show that H3.3 only partially compensated the function of H3mm7; however, the results of RNA-seq analysis in NIH3T3 cells also indicate that H3mm7 has a greater impact on gene expression than H3.3.”

36. Page 11, lines 229-231: The link between this introductory sentence and the following part is not obvious.

Based on the comment, we modified the sentences as follows for a better link.

Original:

“We investigated whether the H3mm7-specific function occurs only after selective recruitment to specific regions induced by recruitment factors such as chaperones specific to H3mm7. To evaluate the effect of histone variant incorporation on changes in chromatin structure, we performed ChIP-seq against GFP, H3K4me3 and H3K27me3 as well as ATAC-seq for each histone variant.”

Updated:

“We investigated whether the H3mm7-specific function occurs only after selective recruitment to specific regions induced by recruitment factors, such as chaperones specific to H3mm7. First, we evaluated if there are unique H3mm7 incorporation sites in NIH3T3 cells ectopically expressing GFP-tagged histone H3 variants. Then, we evaluated histone modifications (H3K4me3 and H3K27me3) at the sites of GFP-tagged histone incorporation as well as chromatin accessibility (ATAC-seq).”

37. Page 11, lines 231-233: In which cells type(s) were this experiments performed?

We appreciate the comment and duly modified the text.

38. Supplementary Fig13 : There is no statistics supporting enrichment of H3.3 and H3mm7 in the mentioned regions.

We performed a binomial test for the peak accumulation in promoter regions where the null hypothesis is that the distribution of peaks was uniform (random) on the genome (e.g., the expected proportion is 1.04% for promoters as indicated in the third lane of the figure). We found that the p -values were 10^{-918} and 10^{-702} for H3.3 and H3mm7 respectively, and concluded that both H3.3-GFP and H3mm7-GFP were specifically enriched in promoter regions.

We added the result of the statistical test to the legend of Supp. Fig. 16 (old Supp. Fig. 13).

39. Fig3C: What is "Weaker than H3.3" and why does it appear at two different places in the graph? Should one of them be "Greater than H3.3" (top-left maybe)? And what is "Enhanced"? What does "Enhanced (negative)" mean? The time point should be indicated in the figure legend.

“Weaker than H3.3” here describes those genes in which the degree of enhancement (in up/downregulation of both expression levels and chromatin accessibility) is smaller than that in the case of the ectopic expression of H3.3-GFP. We agree that the labeling was rather confusing and since these quadrants were not used in the subsequent analysis, we decided to delete these labels.

To clarify the meaning of the remaining labels, we added the following descriptions to the legend.

“Enhanced: up-regulated genes in control NIH3T3 cells over time with higher expression levels and higher accessibility in H3mm7+ cells compared with H3.3+ cells. Enhanced (negative): down-regulated genes in control NIH3T3 cells over time with lower expression levels and lower accessibility in H3mm7+ cells compared with H3.3+ cells.”

The time points are not shown, because the data shows the changes through time (WT vs. H3mm7 for RNA-seq and WT vs. H3mm7 in ATAC-seq) and not at particular time points, i.e., the rate change in RNA-seq and, for ATAC-seq, the gaps between H3mm7 and H3.3 as seen in the Supp. Fig.19 (16 in old Supp.).

40. Fig3D: Do Kif16b and Lmod2 show increased expression upon H3mm7 expression? It does not seem so. Is it normal for Gm20756 and Kif16b expression to show such a "flat" curve across differentiation while they belong to cluster UA?

The reason why the curves look flat is because the vertical axis is scaled to highly expressed genes, such as Gm11884 and Gm12648. If scaled to individual genes, the curves demonstrate the UA pattern as shown below.

41. Page 12, line 258: "Genes in SB (yellow) with [...] where the effect of H3.3 was stronger". Yellow where? Where are these data shown? Effect of H3.3 on what? I do not understand this sentence.

We apologize that we left the reference to the older (before submission) version of the Figure (the yellow points were in the II-quadrant) and we deleted the sentence in question.

In the METHODS section:

42. Page 20, lines 420-421: "[...] and an EGFP cDNA located upstream of the cDNA sequence [...]". The fused proteins should then be referred to as "EGFP-H3.x". However, in several places in the text and in figures, the authors use "H3.3-EGFP" or "H3mm7-EGFP", which would indicate that EGFP is fused to the C-terminal part of the histones (which personally, I would find more logical). The authors should correct this.

This is an important point. We originally analyzed various H3.3 functions in C2C12 cells using N-terminus-tagged cells, as described in the Epigenetics & Chromatin paper, because we also found that N-terminus tags were more efficient at incorporating into chromatin of C2C12 cells. In 2015, we published a paper in Nucleic Acids Res. (Harada et al, 2015), in which we performed a comparative functional study of H3.1 and H3.3. While we understand tagging on histone proteins might interfere with the native function, we found that both C- and N-terminus tags showed comparable effects. Therefore, we chose to use different tags to confirm our observation that H3mm7 caused the rate change of gene expression both in C2C12 cells (N-terminus tag) and NIH3T3 cells (C-terminus tag). We believe that this would not affect our conclusion, but rather support it.

We corrected the order in the labeling (e.g. from "H3-GFP" to "GFP-H3") according to the terminus and added the note to the Methods section.

43. Page 20, lines 423-424: "[...] (2µg plasmid DNA per 6-well plate)". What does that mean? Is it 2 µg per well or something else?

We intended this to mean "2 µg plasmid DNA per well in a 6-well plate" and corrected the sentence.

44. Page 20, line 428: "fluorescence activating cell sorting" is already mentioned before as "FACS" (page 18, line 377). The meaning of FACS should be indicated earlier (page 18) and only the acronym used afterwards.

We appreciate the comment and duly modified the text.

45. Page 21, line 461: the authors should indicate the incubation time for fixation.

We added an explanation as “[...], fixed with 1% paraformaldehyde in PBS for 10 minutes.”

46. Page 25, lines 539-540: “[...] which have three timepoints (UI, 0, 24 and 48 h) and ...”. Isn't it four time points?

It is “four time points” as pointed out. We are grateful for the comment.

Erwan Delbarre

Ph.D.

Reviewer #4 (Remarks to the Author):

The manuscript “Histone H3.3 sub-variant H3mm7 is required for normal skeletal muscle regeneration” by Harada and others described how the structurally more unstable H3.3 variant H3mm7 is incorporated at the same promoters as H3.3, thereby creating more accessible chromatin state, resulting in a general change in expression rate. The authors used a true tour-de-force with a wide range of techniques to probe the function of H3mm7, ranging from generating a H3mm7 KO mouse and C2C12 cell line using CRISPR, immuno-staining, CEL-seq2, ATAC-seq, ChIP-seq, RNA-seq, FRAP, and crystallography.

KO of H3mm7 did not result in up- or down-regulation of specific genes, but rather a more general change in expression rate. This included genes involved in skeletal muscle differentiation. Furthermore, the authors determined that H3mm7 is expressed in earlier stages of skeletal muscle regeneration, most notably quiescent skeletal satellite cells. Indeed, a H3mm7 KO mouse displayed a subtle, but distinct phenotype, namely a delay in muscle regeneration.

Histone variants are thought to evolve by the birth-and-death paradigm. Previously, the authors discovered several H3.3 variants with limited phylogenetic spread, but potentially with functional consequences. Indeed, in this manuscript they delve into the functional implications of one of these novel H3.3 variants, H3mm7. The combination of the KO mouse, various sequencing techniques, and crystallography, a very nice correlation was drawn how subtle structural changes in a minor histone

variant impacts global transcription by creating a more open chromatin structure, which in turn resulted in subtle phenotype.

Overall, we found this manuscript a pleasure to read and, it provides novel insight in how rare histone variants, through relatively subtle means, enhance specific outcomes, in this case, skeletal muscle cell differentiation.

There are only a few minor concerns that, in our view, do not preclude acceptance, but might improve readability.

We are grateful for the reviewers' appreciation of our work and their comments which greatly helped improve the quality of our presentation as well as rectify rather careless mistakes on our part. Point-by-point responses to the comments are given below.

Minor concerns:

1. On lines 131-3, the authors mention that no significant difference was found in total amount of H3. Is there a difference between soluble and chromatin-incorporated H3?

To address this point, we evaluated the total amount of H3 in soluble and chromatin (insoluble) fractions by immunoblotting using an antibody against H3 (H3-C). We found no drastic change in the total H3 level in the chromatin fraction and the soluble H3-C level was decreased only in H3mm7^{-/-} + H3mm7-GFP cells. A further analysis is necessary to understand this phenomenon, but is beyond the scope of this manuscript. While we do not intend to include the following data in the manuscript because of the lack of change of H3-C in whole cell immunoblotting, if the reviewers deem it necessary, we have no objection in adding them.

2. On line 165, the author's mention the "KO effect through time", but no reference is provided.

We apologize for the confusion and the lack of clarity. We meant "KO effect through time" to refer to the second term on the right-hand side of the equation in the manuscript:

Gene expression levels at time t in KO cells = (Wild-type expression level at time t) + (KO effect through time) + (Extra effect at t in KO cells).

Because this topic is discussed in Supp. Fig. 11 (Supp. Fig. 6 in initial version) in more detail, we modified the text as shown below and added Supp. Fig. 11 as a relevant reference.

Original:

"The secondary consequence of the statistical test was that, for all DEGs, the change caused by *H3mm7*^{-/-} could be completely explained by the "KO effect through time", i.e., the effect of *H3mm7*^{-/-} was viewed as a constant rate change (offset in log-scale) through time (Fig. 2a)."

Updated:

"The secondary finding from the statistical test was that, for all DEGs, the change caused by *H3mm7* knockout could be sufficiently explained by the "KO effect through time" term above; i.e., the effect of *H3mm7* knockout was viewed as a constant rate change (offset in log-scale) through time (Fig. 2a, Supp. Fig. 11)."

3. Typo on line 194: Fisher's linear (not liner) discriminant analysis.

We thank the reviewers for catching this and we accordingly corrected the spelling.

4. Line 213, remove 'as' from 'as for C2C12 cells'.

The text was corrected as suggested.

5. Line 214, the authors mention "before infection (UI)". Might the authors have meant induction?

Here we meant “UI” to stand for “uninfected” as we introduced MyoD retrovirally. We clarified it in the text.

6. On line 339, the paragraph starts with a space, rather than a tab.

We appreciate the reviewers’ careful reading. We replaced it with a tab indentation.

7. In Figure 1A, besides H3mm7, H3mm11 appears to have a very similar expression profile, just like the two H3f3a/b genes. Could H3mm11 function synergistically with H3mm7 to change the expression landscape during differentiation, based on tissue expression pattern and that H3mm7 is predominantly found at the promoter and H3mm11 predominantly found at TSS? Maybe the authors could speculate a little in the discussion, to make it a bit more fun for the lay reader?

We are grateful for the reviewers’ suggestion and added a discussion as follows.

“Another histone variant, *H3mm11*, showed an expression pattern closer to that of *H3f3a* than to that of *H3mm7* (Figure 1A). GFP-H3mm11 was detected in abundance in TSS and its gene expression profile was observed to change after differentiation.”

-Yamini Dalal and Daniel Melters

NCI

REVIEWERS' COMMENTS:

Reviewer #1 (Remarks to the Author):

The authors have adequately addressed my previous comments.

Reviewer #2 (Remarks to the Author):

My concerns in the first round review have been addressed with additional data. The revised version of the MS is satisfied with me now.

Reviewer #3 (Remarks to the Author):

The second version of the manuscript "Histone H3.3 sub-variant H3mm7 is required for normal skeletal muscle regeneration" by A. Harada, K. Maehara et al., addresses many of the weaknesses pointed out by this and other reviewers in the first version. I found the answers to my comments satisfactory, in particular regarding statistical analysis matters. Re-phrasing some of the statements comparing H3mm7 and H3.3 roles in gene regulation makes the manuscript stronger in my point of view because this emphasizes the complexity of histone variants in general when it comes to their specific role in chromatin activity. As mentioned in my first review, this paper will be of interest for everyone investigating the different key players of specific chromatin activities, in particular across differentiation and development. This is a manuscript that should be of interest to the readership of Nature Communication.

I only have a few minor points that do not require any further review from my side:

1. Page 5, line 107: "(p=0.006)". I think it should be "(p=0.044)" instead, according to Supplementary Fig. 2a.

2. Page 12, line 244-245: "As a result of k-means clustering [...] six clusters (U, D, S) x (A, B) for C2C12 cells". Shouldn't it be "[...] as for C2C12 cells"? Or "[...] for NIH3T3 cells"?

3. Page 14, line 301, Fig3C and Supplementary Fig.20: I did not find where the "quadrants" were defined. It would help to have I, II, III and IV on Fig3C.

Erwan Delbarre
Ph.D

Reviewer #4 (Remarks to the Author):

The revisions of the manuscript "Histone H3.3 sub-variant H3mm7 is required for normal skeletal muscle regeneration" by Harada et al represent a much-improved version. In our opinion, all concerns that were brought up by all the reviewers have been appropriately addressed. The figures and text are much improved, and much more comprehensible. The addition of the chromatin-bound and soluble H3 in Supplementary data provide clarity between functional H3 and soluble H3. We believe this article is suitable for publication.

-Dalal & Melters, NCI/NIH

RESPONSE TO REVIEWERS' COMMENTS:

Reviewer #1 (Remarks to the Author):

The authors have adequately addressed my previous comments.

We are grateful for all the comments which significantly bettered the quality of the manuscript.

Reviewer #2 (Remarks to the Author):

My concerns in the first round review have been addressed with additional data. The revised version of the MS is satisfied with me now.

We greatly appreciated the concerns which Reviewer #2 expressed, which were thought-provoking and helped improve our manuscript as well as our understanding of the material.

Reviewer #3 (Remarks to the Author):

The second version of the manuscript “Histone H3.3 sub-variant H3mm7 is required for normal skeletal muscle regeneration” by A. Harada, K. Maehara et al., addresses many of the weaknesses pointed out by this and other reviewers in the first version. I found the answers to my comments satisfactory, in particular regarding statistical analysis matters. Re-phrasing some of the statements comparing H3mm7 and H3.3 roles in gene regulation makes the manuscript stronger in my point of view because this emphasizes the complexity of histone variants in general when it comes to their specific role in chromatin activity. As mentioned in my first review, this paper will be of interest for everyone investigating the different key players of specific chromatin activities, in particular across differentiation and development. This is a manuscript that should be of interest to the readership of Nature Communication.

We are thankful for the reviewer’s comments and suggestions which rendered the manuscript much more readable.

I only have a few minor points that do not require any further review from my side:

1. Page 5, line 107: "(p=0.006)". I think it should be "(p=0.044)" instead, according to Supplementary Fig. 2a.

We thank Reviewer #3 for catching this mistake which would have gone otherwise unnoticed. We accordingly corrected the value.

2. Page 12, line 244-245: "As a result of k-means clustering [...] six clusters (U, D, S) x (A, B) for C2C12 cells". Shouldn't it be "[...] as for C2C12 cells"? Or "[...] for NIH3T3 cells"?

The reviewer is again correct and we are grateful. We replaced “C2C12” with “NIH3T3.”

3. Page 14, line 301, Fig3C and Supplementary Fig.20: I did not find where the "quadrants" were defined. It would help to have I, II, III and IV on Fig3C.

We followed the suggestion and added a graphical description of the quadrants in Fig. 3C.

Erwan Delbarre
Ph.D

Reviewer #4 (Remarks to the Author):

The revisions of the manuscript “Histone H3.3 sub-variant H3mm7 is required for normal skeletal muscle regeneration” by Harada et al represent a much-improved version. In our opinion, all concerns that were brought up by all the reviewers have been appropriately addressed. The figures and text are much improved, and much more comprehensible. The addition of the chromatin-bound and soluble H3 in Supplementary data provide clarity between functional H3 and soluble H3. We believe this article is suitable for publication.

-Dalal & Melters, NCI/NIH

We appreciate the reviewers' comments; following their suggestions, the said data have now been incorporated into Supplementary Figure 7b.